# LAMP: DATA-EFFICIENT LINEAR AFFINE WEIGHT-SPACE MODELS FOR PARAMETER-CONTROLLED 3D SHAPE GENERATION AND EXTRAPOLATION

## ABSTRACT

Generating high-fidelity 3D geometries that satisfy specific parameter constraints has broad applications in design and engineering. However, current methods typically rely on large training datasets and struggle with controllability and generalization beyond the training distributions. To overcome these limitations, we introduce LAMP (Linear Affine Mixing of Parametric shapes), a data-efficient framework for controllable and interpretable 3D generation. LAMP first aligns signed distance function (SDF) decoders by overfitting each exemplar from a shared initialization, then synthesizes new geometries by solving a parameter-constrained mixing problem in the aligned weight space. To ensure robustness, we further propose a safety metric that detects geometry validity via linearity mismatch. We evaluate LAMP on two 3D parametric benchmarks: DrivAerNet++ and Blended-Net. We found that LAMP enables (i) controlled interpolation within bounds with as few as 100 samples, (ii) safe extrapolation by up to 100% parameter difference beyond training ranges, (iii) physics performance-guided optimization under fixed parameters. LAMP significantly outperforms conditional autoencoder and Deep Network Interpolation (DNI) baselines in both extrapolation and data efficiency. Our results demonstrate that LAMP advances controllable, data-efficient, and safe 3D generation for design exploration, dataset generation, and performance-driven optimization.

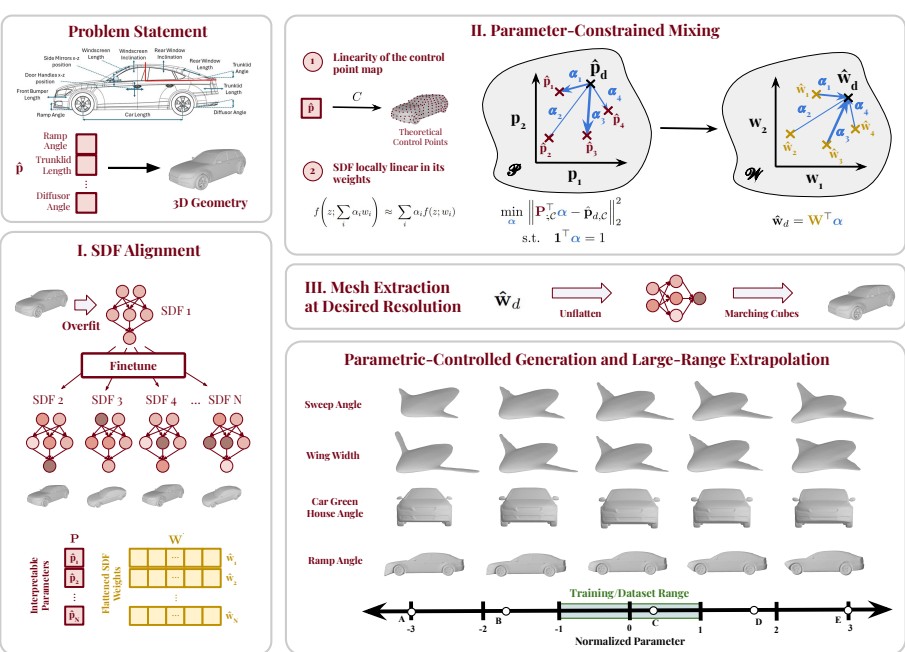

Figure 1: Overview of LAMP: (I) aligned SDF weight space construction, (II) parameter-constrained mixing, and (III) mesh extraction, enabling parametric control and large-range extrapolation.

## 1 INTRODUCTION

Engineering design applications often require generating 3D geometries that satisfy explicit, human-interpretable parameters (e.g., aerodynamic drag, roof height, and ramp angle). However, these applications are often data-scarce and require exploration beyond the limited span of available exemplars and parameter values unseen in the training distribution (e.g., generating a 3D car shape with a much larger ramp angle compared to known cars in the dataset). This opens up a fundamental challenge of how we can enable controllability and interpretability beyond training distributions in generative models across many domains, such as images, 3D shapes, text, and audio.

Many existing methods achieve control by learning a shared latent space and then traversing or disentangling latent dimensions to align with semantic or parametric factors (Vahdat et al., 2022; Zhao et al., 2023; Morita et al., 2024; Xiang et al., 2025). While powerful, latent approaches require a single generator to capture all variability, achieving clean, interpretable directions often proves difficult. The problem is particularly challenging in 3D shape generation, as most 3D generative models—whether voxel, point cloud, mesh, or implicit—rely on large datasets and provide few guarantees about parameter control or generalization beyond the training distribution.

We introduce **LAMP** (Linear Affine Mixing of Parametric shapes), a data-efficient framework for parameter-controlled 3D mesh generation. LAMP aligns signed distance function (SDF) decoders by overfitting each exemplar from a shared initialization, producing a consistent weight-space basis tied to design parameters. Given a target specification, we solve a constrained mixing problem to obtain coefficients, linearly combine exemplar weights, and decode a mesh. In other words, we cast controllable generation as parameter-space affine mixing across exemplar-specific networks.

This formulation leverages two simple but powerful observations: (i) interpretable attributes in many domains often combine approximately linearly in a suitable basis (e.g., control-point deformations in geometry), and (ii) neural decoders behave approximately linearly in their weights in a local regime around a common initialization. Together, these properties allow us to replace complex disentanglement or conditioning strategies with a lightweight linear solve and direct parameter mixing. We show that this method allows extrapolation beyond the training data and requires few samples.

To demonstrate reliability in meeting parameter constraints, we also introduce a method for direct *linearity-mismatch* safety check: for sampled 3D points, we compare the mixed decoder's output with the linear combination of individual SDFs, accepting generations only if the mean error is below 0.01. Constraint compliance is evaluated using a mesh-based surrogate that predicts parameters from PointNet embeddings with $R^2 > 0.9$ Qi et al. (2017), complemented by direct geometric measurements on decoded meshes. This combination provides both flexibility and accountability in design generation.

We validate LAMP on two parametric benchmarks, DrivAerNet++ and BlendedNet, across four case studies: (i) controlled interpolation within dataset bounds, (ii) safe extrapolation up to 100% beyond training ranges, and (iii) optimization for aerodynamic properties under fixed parameters. Together, these results demonstrate that LAMP enables controllable, data-efficient, and safe mesh generation for design exploration and performance-driven optimization.

LAMP thus bridges the gap between data-efficient generative modeling and practical engineering design. By combining aligned SDF weight spaces, provably safe mixing, and mesh-based surrogate validation, our framework provides a principled foundation for parameter-controlled 3D mesh generation. This enables not only faithful reproduction of in-distribution designs but also safe exploration of new configurations, including extrapolated and performance-optimized geometries. Ultimately, LAMP offers a scalable approach to transform a few annotated exemplars into a versatile design engine for constrained yet creative geometry synthesis.

More specifically, our contributions can be summarized as follows:

- **LAMP Method.** We propose LAMP, a data-efficient framework for parameter-controlled 3D mesh generation that aligns exemplar-specific SDF decoders to create a shared weight space and synthesizes new shapes through affine mixing from it.

- **Safety Metric.** We introduce a linearity-mismatch metric that certifies whether interpolated or extrapolated weight-space combinations by LAMP remain geometrically valid.

- **Engineering Applications.** We demonstrate LAMP on two benchmarks of aerodynamic 3D design (DrivAerNet cars and BlendedNet aircraft), showing controlled interpolation, large-range extrapolation (up to 100% beyond dataset bounds), and performance-driven optimization.

## 2 RELATED WORK

**Generative Models for 3D Shapes.** A wide range of 3D generative models have been proposed, spanning voxel grids Wu & et al. (2016), point clouds Achlioptas & et al. (2018), meshes Groueix & et al. (2018), and neural implicit representations such as signed distance functions (SDFs) Park et al. (2019); Chibane et al. (2020). Neural SDFs capture high-resolution geometry and have been applied to generation and reconstruction Chen & et al. (2022) and reconstruction Atzmon & Lipman (2020). Recent diffusion-based 3D generators operate on implicit or latent representations, including LION, GET3D, Diffusion-SDF, SDFusion, and SALAD Zeng et al. (2022); Gao et al. (2022); Chou et al. (2023); Cheng et al. (2023); Koo et al. (2023). However, these methods typically assume abundant training data and lack explicit mechanisms for parameter-constrained generation or safe extrapolation. More recently, HyperDiffusion Erkoç et al. (2023) modeled the distribution of overfit implicit networks directly in weight space, sampling new fields via diffusion. These works treat networks themselves as data points in parameter space, but focus on unconditional sampling or learned meta-combination rather than explicit affine mixing under interpretable constraints.

**Controllable and Conditional Generation.** Efforts to introduce control often rely on conditioning on labels or attributes Gao et al. (2019); Niemeyer et al. (2020), or on parametric templates derived from CAD data Yumer & Mitra (2016); Wang et al. (2022). Diffusion models have recently been adapted for class-conditional and partially conditional shape generation Chen & et al. (2023); Liu & et al. (2023). Multimodal conditioning has also been explored for controllable 3D generation, e.g., CLIP-Forge and Michelangelo Sanghi et al. (2022); Zhao et al. (2023). However, these methods rarely support precise parameter specification (e.g., generating a car body with fixed ramp angle and width) and are not designed for data-efficient regimes. Our approach complements this line of work by enabling direct control through interpretable parameters.

**Weight-Space Interpolation and Model Merging** A few works have shown that linear operations in weight space can produce coherent outputs. Deep Network Interpolation (DNI) Wang et al. (2019) demonstrated smooth visual transitions by averaging parameters of two correlated image translation networks. In classification, model soups Wortsman et al. (2022) average fine-tuned networks to improve robustness. In generative modeling, researchers have merged GANs trained on different categories to yield hybrid semantics Avrahami et al. (2022), and also commonly combine diffusion models by interpolating or adding weight deltas to merge styles Biggs et al. (2024). These approaches confirm that weight-space mixing can yield meaningful interpolations, but typically operate on pairs of models and lack mechanisms for interpretable, constraint-driven control.

**Parametric Design and Engineering Constraints.** Engineering design relies heavily on parametric modeling, where small sets of interpretable variables govern global shape Seff & et al. (2020); Wang et al. (2022); Du & et al. (2023). While recent learning-based CAD systems leverage symbolic histories or constraint graphs, they often require large, structured datasets. By contrast, we target the data-efficient setting where only meshes and parameter annotations are available. LAMP directly links parameters to mesh geometry via aligned SDF weight spaces, and enforces validity through a linearity-mismatch safety metric and mesh-based surrogate checks.

**Shape Interpolation and Extrapolation.** Latent-space interpolation has been widely explored in autoencoders Achlioptas & et al. (2018) and implicit representations Park et al. (2019), but these spaces are often not semantically aligned, leading to unrealistic interpolations or invalid extrapolations. Our method leverages affine mixing of aligned SDF weights, which—combined with the linearity-mismatch criterion—ensures that interpolated or extrapolated meshes remain geometrically consistent and satisfy parametric constraints.

**Position of This Work.** Our approach bridges these threads. Like HyperDiffusion, we treat overfit exemplar networks as aligned points in parameter space, but instead of learning a generative model

over them, we provide direct, interpretable control by solving for mixing coefficients in parameter space and applying them in weight space. Unlike latent traversal or disentanglement, we do not rely on a single model to encode all variation. And unlike prior weight interpolation methods, we generalize beyond two-model blends to a bank of exemplars, enabling constraint-driven, multi-way affine mixing. To our knowledge, this is the first work to formulate controllable generation as parameter-space affine mixing of aligned exemplar networks.

# 3 METHOD

We present **LAMP**, a data-efficient framework for controllable 3D mesh generation that can safely interpolate and extrapolate in parameter space. LAMP (i) constructs an *aligned* weight-space basis by overfitting signed distance function (SDF) networks to a small set of 3D shapes, (ii) solves a parameter-constrained *mixing* problem to synthesize new SDF weights and decode meshes, and (iii) evaluates reliability using a linearity-based safety metric and a surrogate that predicts parameters directly from generated meshes.

**Problem Setup and SDF Weight Space**   We are given $N$ exemplars, each with mesh $\mathcal{M}_i$ and parameter vector $\mathbf{p}_i \in \mathbb{R}^d$ (e.g., length, width, roof height, ramp angle). For every design, we overfit an SDF network $f_{\theta_i}$, starting from a shared initialization at the mean design. This yields weights $\mathbf{w}_i = \theta_i \in \mathbb{R}^D$ that live in an approximately *aligned* weight space. Stacking rows gives

$$\mathbf{P} \in \mathbb{R}^{N \times d}, \qquad \mathbf{W} \in \mathbb{R}^{N \times D}.$$

An arbitrary weight vector $\mathbf{w}$ is decoded into a mesh $\mathcal{M} = \mathrm{Decode}(\mathbf{w})$ by evaluating the zero-level set of the SDF distribution on a dense voxel grid at the desired resolution, and extracting the isosurface using the marching cubes algorithm Lorensen & Cline (1998).

**Parameter-Constrained Mixing**   Given a target parameter vector $\mathbf{p}_d$ with a constrained index set $\mathcal{C}$, we solve the following optimization problem for mixing coefficients $\alpha$:

$$\min_{\alpha} \left\| \mathbf{P}_{:,\mathcal{C}}^{\top} \alpha - \mathbf{p}_{d,\mathcal{C}} \right\|_2^2 \quad \text{s.t.} \quad \mathbf{1}^{\top} \alpha = 1. \tag{1}$$

The synthesized weights and decoded mesh are

$$\mathbf{w}_d = \mathbf{W}^{\top} \alpha, \qquad \mathcal{M}_d = \mathrm{Decode}(\mathbf{w}_d). \tag{2}$$

Negative $\alpha$ is allowed, enabling extrapolation beyond dataset bounds.

**Theoretical Justification of Mixing**   Our framework builds on two key assumptions:

(A1) *Linearity of the control-point map.* Each mesh $\mathcal{M}_p$ can be described by a set of geometric control points $C(p) \in \mathbb{R}^{n \times 3}$, such as spline knots or characteristic vertices that define the surface. We assume the map from parameters to control points is linear:

$$C\left( \sum_i \alpha_i p_i \right) = \sum_i \alpha_i C(p_i).$$

This reflects how most engineering deformations are modeled: affine transformations (translation, scaling, stretching), spline coefficient adjustments, or superpositions of independent deformations. Even when nonlinear parameterizations are used (e.g., quadratic variations in thickness or curvature), they can often be re-expressed in a linear basis of control-point coefficients (see Appendix A).

(A2) *Local linearity in SDF weights.* For fixed input $z$, the decoder $f(z; w)$ is approximately linear in $w$ in a neighborhood of a reference $w_0$:

$$f\left( z; \sum_i \alpha_i w_i \right) \approx \sum_i \alpha_i f(z; w_i),$$

with error $O(\max_i \|w_i - w_0\|^2)$. (Proof in Appendix C)

Under (A1)–(A2), interpolating weights $\hat{w}_\alpha = \mathbf{W}^{\top} \alpha$ produces an SDF close to $\mathrm{SDF}(C(\hat{p}_\alpha))$ with $\hat{p}_\alpha = \mathbf{P}^{\top} \alpha$, ensuring that mixing in weight space corresponds to faithful geometric interpolation and extrapolation. (Proof in Appendix B)

**Safety Metric: Linearity Mismatch**   We additionally quantify whether affine mixing remains in a valid linear regime (A2). For $N_z$ sampled 3D coordinates $\{z\}$, we compute

$$\frac{1}{N_z} \sum_z \left| f(z; \sum_i \alpha_i w_i) - \sum_i \alpha_i f(z; w_i) \right|.$$

A mesh is accepted if this mismatch is below $\epsilon$. This provides a quantitative safety threshold: low mismatch implies faithful linear mixing, while high mismatch indicates collapse (Fig. 5).

## 4   EXPERIMENTS, RESULTS, AND DISCUSSION

We evaluate LAMP against two representative baselines for parameter-controlled 3D generation:

**DNI (Deep Network Interpolation)**: a learned model mapping design parameters directly to SDF decoder weights Wang et al. (2019). **AE-LPA (Autoencoder with Latent-Parameter Alignment)**: an autoencoder trained to reconstruct SDF weights with its latent subspace linearly aligned to design parameters Jain et al. (2021).

We benchmark these methods on two recent parametric datasets (Appendix J): **DrivAerNet++**: a large-scale multimodal car dataset with $\sim 8{,}000$ distinct geometries, each annotated with 26 interpretable design parameters, high-resolution meshes, and CFD-based aerodynamic coefficients. Elrefaie et al. (2024) **BlendedNet**: a blended wing-body (BWB) aircraft dataset with 999 geometries, each simulated under 9 flight conditions, and annotated with planform parameters such as chord-length ratios, spanwise widths, and sweep angles. Sung et al. (2025) For brevity, we focus on DrivAerNet++ in the main paper and report full BlendedNet results in Appendix D.

**Evaluation Metrics**   For in-dataset generation (when the target mesh is available), we evaluate: **Chamfer Distance (CD)** and **Intersection-over-Union (IoU)** between generated and reference meshes (see Appendix G). **Parameter Error:** mean absolute error (MAE) between target parameters and surrogate-predicted parameters for the generated mesh. For out-of-distribution extrapolation (no ground-truth mesh), we evaluate: **Parameter Fidelity:** surrogate-predicted MAE and $R^2$ between target parameters $p_d$ and inferred $\hat{p}$, **Distributional Similarity:** Minimum Matching Distance (MMD) between generated shapes and a reference set of geometries. (see Appendix G).

**Constraint Compliance via Mesh-Based Surrogates**   To assess whether generated shapes respect design-parameter constraints, we employ a mesh-based surrogate model trained to predict interpretable parameters (e.g., geometric or performance attributes). We compute fixed, randomly initialized PointNet embeddings for each decoded mesh Amid et al. (2022), and fit a LASSO regressor Tibshirani (1996) to map embeddings to physical parameters. Despite using an untrained encoder, the surrogate consistently achieves $R^2 > 0.9$ on held-out test sets, providing a robust parameter validator. When possible, we further cross-check compliance through *direct geometric measurements* of the meshes (details in Appendix H).

**Interpolation within Dataset Range**   We evaluate interpolation by reconstructing meshes from randomly sampled dataset examples that fall within the parameter range but are excluded from training. As shown in Table 1, LAMP achieves the best performance across Chamfer Distance, IoU, and parameter error. Notably, with only 100 samples, it surpasses AE-LPA trained on 1000 samples, demonstrating strong sample efficiency in low-data regimes.

Table 1: Interpolation results on DrivAerNet++. Comparison of Chamfer Distance (CD), IoU, and parameter error (MAE) across methods.

| Method | # Samples | CD $\downarrow$ | IoU $\uparrow$ (%) | MAE $\downarrow$ |
|---|---|---|---|---|
| DNI | 100 | 0.0118 | 97.12 | 0.0015 |
| AE-LPA | 100 | 0.0181 | 88.21 | 0.0025 |
| AE-LPA | 1000 | 0.0144 | 92.63 | 0.0020 |
| **LAMP (Ours)** | 100 | **0.0117** | **97.24** | **0.0014** |

**Extrapolation within Dataset Range**  We next evaluate extrapolation slightly beyond the training set. Training samples are drawn from a centered 50% interval of the parameter range, and evaluation is performed on cars with parameter values outside that interval. Figure 8 illustrates the *single-parameter* case, showing extrapolation along front bumper curvature. While DNI—the strongest baseline within the dataset range—begins to drift outside the training span, LAMP maintains smooth, parameter-consistent geometry.

Table 2 reports quantitative results. For single-parameter extrapolation, LAMP improves Chamfer Distance, IoU, and surrogate error relative to DNI. The advantage grows in the *multi-parameter* setting, where three random parameters are simultaneously extrapolated outside a 60% centered range: LAMP reduces surrogate MAE by 20–30%, showing greater robustness under multiple constraints.

Table 2: Extrapolation within dataset range (DrivAerNet++). LAMP outperforms DNI in both single- and multi-parameter settings, with especially large gains when multiple parameters are extrapolated simultaneously.

| Method | Single Parameter | | | Multi-Parameter (3) | | |
|---|---|---|---|---|---|---|
| | CD ↓ | IoU ↑ | MAE ↓ | CD ↓ | IoU ↑ | MAE ↓ |
| DNI | 0.0129 | 95.32 | 0.098 | 0.0139 | 94.28 | 0.186 |
| **LAMP (Ours)** | **0.0126** | **95.75** | **0.077** | **0.0130** | **95.29** | **0.144** |

**Large-Range Extrapolation Beyond the Dataset Bounds**  We next evaluate extrapolation far outside the training span, extending parameters up to $\pm 100\%$ beyond the dataset limits (i.e., three times the original parameter range). This task is challenging, as models must generate plausible geometries without support from nearby training examples. We consider two extrapolation settings in our experimental setup:

1. **Single-Parameter Extrapolation.** For each parameter, we sweep across the extrapolated range ($3\times$ the dataset span) using 10 uniformly sampled target values, while allowing all other parameters to vary freely. This setup evaluates whether models sustain smooth and consistent shape evolution along a single direction of variation (Figs. 3, 2).

2. **Multi-Parameter Extrapolation (4D).** We repeat 100 trials where four parameters are randomly selected and set outside the dataset range (up to 50% extrapolation, i.e., $2\times$ the span). Each method generates meshes under these conditions, which are then converted to point clouds, embedded with a fixed PointNet encoder, and visualized in 2D via multidimensional scaling (MDS). This reveals both fidelity and diversity of extrapolated generations (Fig. 4).

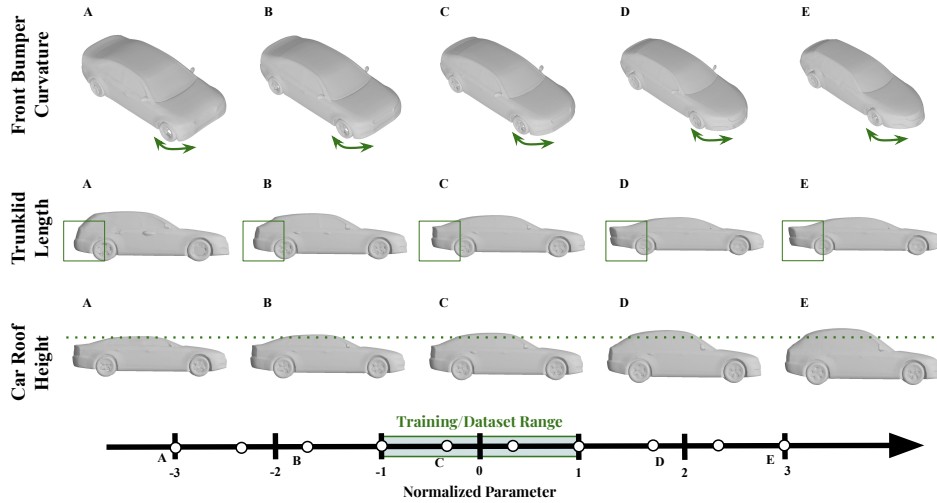

Figure 2: Single-parameter extrapolation showing LAMP's smooth, plausible geometries.

Table 3: Large-range extrapolation (DrivAerNet++). LAMP sustains high fidelity ($R^2 > 0.86$), while DNI collapses ($R^2 < 0$).

| Method | Single Parameter | | | Multi-Parameter (4) | | |
|---|---|---|---|---|---|---|
| | MMD ↓ | MAE ↓ | $R^2$ ↑ | MMD ↓ | MAE ↓ | $R^2$ ↑ |
| DNI | 0.043 | 0.705 | 0.143 | 0.060 | 1.313 | -5.768 |
| AE-LPA | 0.031 | 0.405 | 0.750 | 0.030 | 0.420 | 0.685 |
| **LAMP (Ours)** | **0.030** | **0.247** | **0.902** | **0.030** | **0.324** | **0.867** |

**Results.**  Quantitative results are reported in Table 3. For single-parameter extrapolation, LAMP reduces parameter error by more than 40% compared to DNI and achieves $R^2 = 0.90$ versus $R^2 = 0.14$ for DNI. In the four-parameter case, DNI collapses completely ($R^2 < 0$), AE-LPA remains confined to the convex hull of the dataset, while LAMP sustains high fidelity ($R^2 = 0.87$) with low MMD and surrogate error.

Figures 2 and 3 illustrate the single-parameter sweeps: DNI collapses outside the training range and AE-LPA undershoots, whereas LAMP produces smooth, parameter-consistent variations that remain geometrically valid across the entire sweep. In the more challenging four-parameter extrapolation (Fig. 4), DNI collapses to invalid meshes and scatters randomly in embedding space, AE-LPA stays trapped in the dataset's convex hull with low diversity, while LAMP extrapolates beyond the convex hull and generates high-fidelity meshes in previously unobserved regions. This shows that LAMP can be used for dataset augmentation and controlled generation from a few samples.

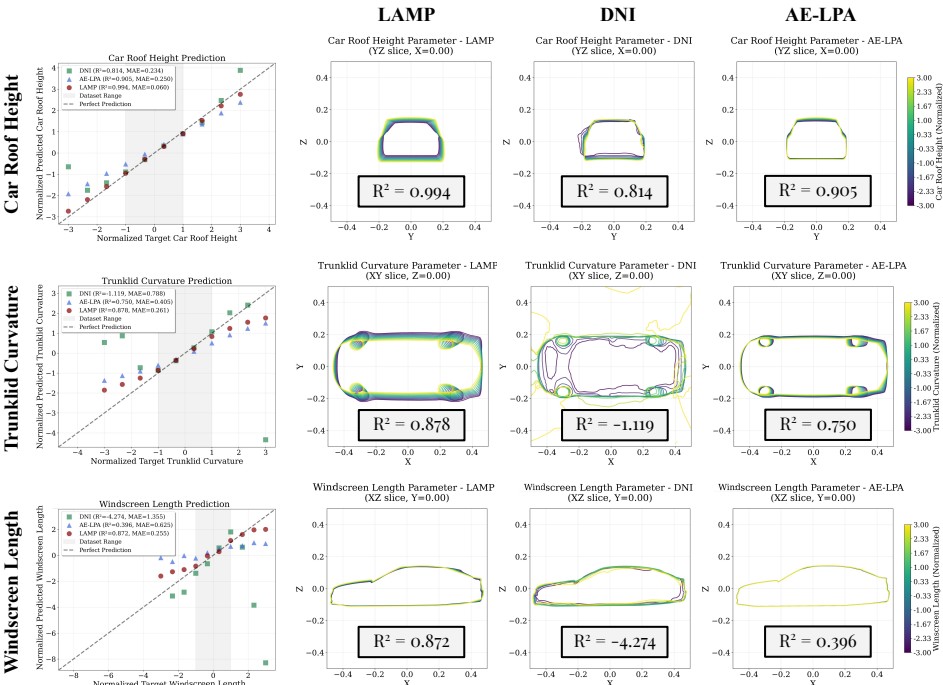

Figure 3: Single-parameter extrapolation beyond the dataset range, with all other parameters allowed to vary. Left: surrogate-predicted vs. target parameters. Right: decoded cross-sections. LAMP extrapolates smoothly, while DNI collapses and AE-LPA fails to reach the expected parameter range.

**Challenges and Safety in Extrapolation with Limited Data**  A major challenge of large-range extrapolation is validating the plausibility of generated geometries when training data are scarce. With only 100 samples, mesh-based surrogates cannot be trained reliably to evaluate out-of-distribution designs. In such low-data regimes, models may occasionally produce collapsed or implausible shapes, especially when extrapolating far beyond the dataset span.

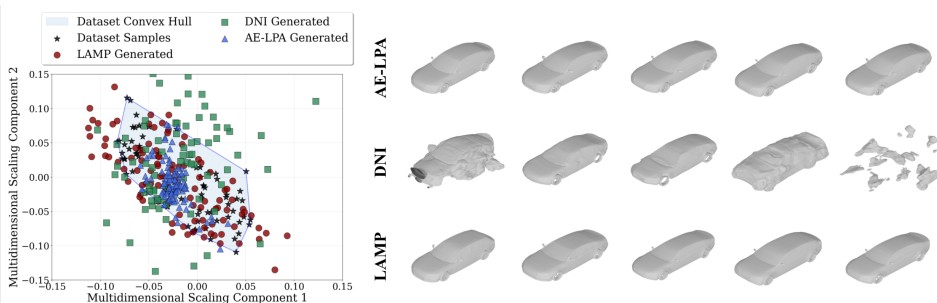

Figure 4: Four-parameter extrapolation. Left: distribution of generated meshes in a 2D point cloud embedding. Right: decoded examples. LAMP remains within plausible regions, DNI collapses to invalid meshes, and AE-LPA remains stuck in the dataset convex hull, lacking diversity.

To address this, we introduce a *linearity-mismatch safety metric* (Sec. 3), which quantifies whether affine weight mixing remains locally valid in SDF space. Unlike surrogate-based validation, this metric is lightweight and data-independent, enabling it to flag unsafe generations even when labeled training data are unavailable. As shown in Fig. 5, failure cases (f) arise precisely when the mismatch score exceeds a threshold.

We validate this metric against a human-annotated dataset of valid and invalid meshes (Appendix I). The results show excellent discriminative power (ROC AUC = 0.989, PR AUC = 0.990), and $\epsilon = 0.01$ emerges as a reliable threshold for separating valid from invalid generations. Additional examples of failure cases and validation analysis are provided in Figs. 18–19.

Another limitation is that we were only able to hard-code the measurement of car length, as explained in Appendix H, achieving $R^2 = 0.999$ under $\pm 100\%$ extrapolation sweeps. In contrast, other parameters were far more difficult to hard-code due to the complexity of their deformations across different car geometries and the absence of reliable methods for accurate measurement.

Finally, we study how reliability scales with data availability (Appendix E). As shown in Table 7, increasing the training set size improves both predictive accuracy ($R^2$, MAE) and the mean safe extrapolation range, which grows from $\sim 145\%$ with 10 samples to over $400\%$ with 1000 samples before saturating. This ablation highlights both the limitations of extrapolation in extremely low-data regimes and the safety metric's role in flagging unreliable extrapolations.

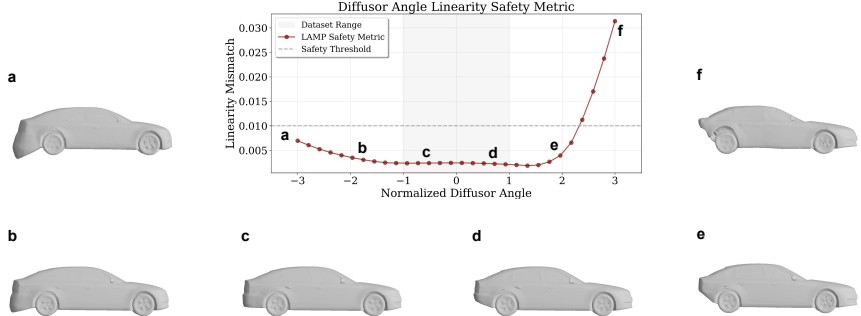

Figure 5: Linearity-mismatch safety metric for diffuser angle extrapolation. Failures (e.g., sample f) occur when the metric exceeds the threshold. See Appendix I for more.

**Performance-Driven Optimization**   Beyond geometric parameters, we also test whether LAMP can enable *performance-based control*, where the goal is to optimize aerodynamic properties while constraining selected physical parameters. Specifically, we sample 100 random test examples from DrivAerNet++ outside the training set. For each example, we decay the drag coefficient ($C_d$) by $10\%$ and select a random subset of physical parameters to be constrained to their original values, while treating the decayed $C_d$ as an additional desired parameter. We then solve for mixing coefficients $\alpha$ that jointly satisfy these constraints.

To validate the results, we use the mesh-based surrogate (Appendix H) to predict both physical parameters and drag coefficients from the generated meshes. The surrogate predictions are compared to ground-truth values, with predicted vs. target plots provided in the Appendix. We evaluate two objectives: (i) *parameter fidelity*, i.e. how closely the generated meshes respect the selected physical parameter constraints, and (ii) *drag fidelity*, i.e. how accurately the achieved reduction matches the 10% target. Here, *decay MAE* denotes the mean absolute error between the desired 10% decay and the observed (predicted) decay, averaged across all samples.

Table 4: Performance-driven optimization on DrivAerNet++ for a 10% drag reduction target. LAMP achieves the best balance between parameter fidelity and aerodynamic performance.

| Method | Physical Parameters | | Drag Coefficient | | |
|---|---|---|---|---|---|
| | MAE $\downarrow$ | $R^2 \uparrow$ | Decay MAE $\downarrow$ (%) | MAE $\downarrow$ | $R^2 \uparrow$ |
| DNI | 0.810 | -0.184 | 10.2 | 0.333 | -8.917 |
| AE-LPA | 0.161 | 0.797 | 5.2 | 0.146 | 0.297 |
| **LAMP (Ours)** | **0.087** | **0.938** | **2.7** | **0.121** | **0.792** |

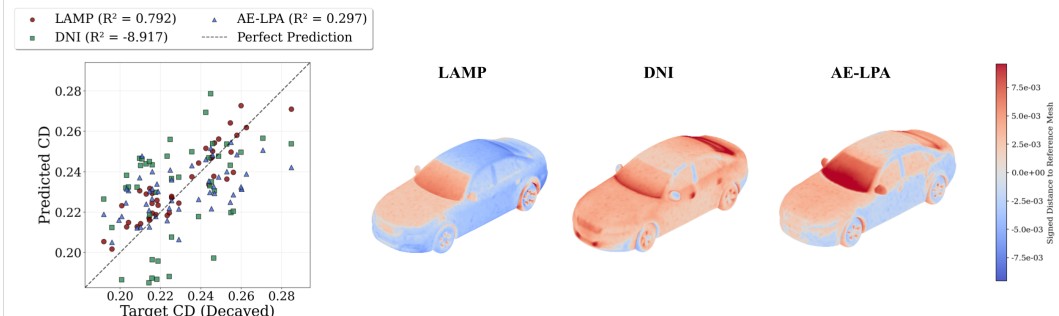

Figure 6: Performance-driven drag optimization on DrivAerNet++. Left: target vs. predicted drag reduction for LAMP, DNI, and AE-LPA. Right: error heatmaps relative to the reference mesh. LAMP achieves accurate prediction and a physically interpretable modification (flattened windscreen), while DNI and AE-LPA fail to produce aerodynamically meaningful changes.

**Results.** Figure 6 and Table 4 summarize the outcomes. The scatter plot confirms that LAMP aligns strongly with the target drag decay ($R^2 = 0.792$), while DNI diverges completely ($R^2 < 0$) and AE-LPA shows weaker correlation. Error heatmaps highlight the geometric changes driving drag reduction: LAMP produces a visibly flatter windscreen angle, reducing flow separation and lowering drag, whereas DNI and AE-LPA introduce noisy or less interpretable deformations.

Quantitatively, DNI fails to satisfy both aerodynamic and parametric constraints, with large parameter drift (MAE = 0.810) and unstable drag predictions ($R^2 = -8.917$). AE-LPA maintains moderate parameter fidelity ($R^2 = 0.797$) but overshoots the decay target (decay MAE = 5.2%). In contrast, LAMP achieves the best trade-off: parameter fidelity improves to MAE = 0.087 with $R^2 = 0.938$, and drag reduction error is reduced to just 2.7%, while maintaining the strongest correlation for drag. Together, these results show that LAMP not only respects parameter constraints but also identifies physically meaningful pathways for aerodynamic optimization.

## 5 LIMITATIONS AND FUTURE WORK

While LAMP enables data-efficient and interpretable parameter-controlled 3D generation, it has several limitations. First, the method assumes that all exemplars share a common topological structure and satisfy the linear control-point model in Assumption A1; using geometries with differing topology or strongly nonlinear deformations would violate this assumption and invalidate affine mixing. Second, reliable extrapolation depends on remaining within the locally linear weight-space regime, which may break under large extrapolation or insufficient exemplar coverage. Finally, LAMP may fail when target performance objectives and physical parameters impose conflicting requirements, yielding combinations that are not jointly realizable within the exemplar set. Future work will aim to

extend this framework to support multiple topological families, integrate physics-aware constraints, and develop multi-modal conditioning pipelines.

## 6 CONCLUSION

We presented LAMP, a data-efficient framework for parameter-controlled 3D mesh generation that leverages affine mixing in aligned SDF weight spaces and a linearity-based safety metric. Experiments on DrivAerNet++ and BlendedNet show that LAMP outperforms conditional autoencoders and DNI across interpolation, large-range extrapolation, and performance-guided optimization, achieving reliable control with as few as 100 exemplars. The safety score provides a principled safeguard in low-data regimes, addressing a key challenge for robust generalization. LAMP advances the goal of controllable, efficient, and verifiable 3D generation, common in engineering applications.

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

TABLE OF CONTENTS FOR APPENDICES

## A    ON THE LINEARITY OF THE CONTROL-POINT MAP

Our theoretical justification in §3 requires assumption (A1), i.e., that the control points defining a mesh are a linear function of design parameters. Here we explain why this assumption is natural and broadly applicable.

**Affine deformations.**    If a mesh is transformed by translation, scaling, or uniform stretching along a coordinate axis, then each control point is exactly a linear function of the corresponding parameter. For example, increasing the wing span of an aircraft by $\Delta s$ simply adds $\Delta s$ to the $x$-coordinates of the wingtip control points.

**Parameterized curves and surfaces.**    For many design families, parameters control polynomial or spline coefficients. Since a spline curve is itself a linear combination of control points, perturbing these coefficients changes the embedding linearly in parameter space. Even nonlinear geometric trends (e.g., quadratic camber variation) can be re-expressed in a linear basis of coefficients.

**Superposition of deformations.**    When multiple independent deformations (length, width, rotation about an axis) are applied, the resulting control-point positions are affine functions of all parameters. Thus, any convex combination of parameter vectors yields a convex combination of control-point sets, consistent with (A1).

**Coverage of practical deformations.**    Most engineering shape variation in engineering practice can be decomposed into linear control-point operations: extrusion height, lofting length, angle of attack, or wheelbase translation are all captured. More exotic nonlinear changes (e.g., tree-like topological branching) violate (A1) but are outside the scope of our controlled parametric families.

**Takeaway.** Assumption (A1) is not an artificial simplification but instead reflects how engineering models are actually parameterized: the majority of mesh variations of interest in engineering design are affine in a suitable control-point basis. This ensures that our interpolation scheme faithfully reproduces the geometry implied by parameter mixing in nearly all practical scenarios.

## B    THEORETICAL JUSTIFICATION OF SDF WEIGHT INTERPOLATION

**Theorem.** Let $p_1, \ldots, p_N \in \mathbb{R}^d$ be parameter vectors defining meshes $\mathcal{M}_{p_i}$ via control points $C(p_i) \in \mathbb{R}^{n \times 3}$, where the map $x \mapsto C(x)$ is linear. Let $w_i \in \mathbb{R}^m$ denote the weights of an MLP SDF decoder $f(z; w)$ overfit to the mesh $\mathcal{M}_{p_i}$, and trained from a shared initialization. Suppose that:

(1) Control point interpolation is linear:

$$C\left(\sum_{i=1}^{N} \alpha_i p_i\right) = \sum_{i=1}^{N} \alpha_i C(p_i), \quad \sum_{i=1}^{N} \alpha_i = 1.$$

(2) Each MLP decoder satisfies:

$$f(z; w_i) \approx \text{SDF}(z; C(p_i)) := d_i(z),$$

where $d_i(z)$ denotes the signed distance from a queried location $z$ to mesh $\mathcal{M}_{p_i}$.

(3) The decoder $f(z; w)$ is locally linear in weights $w$ for fixed input $z$:

$$f\left(z; \sum_{i=1}^{N} \alpha_i w_i\right) \approx \sum_{i=1}^{N} \alpha_i f(z; w_i),$$

with error bounded by $O(\max_i \|w_i - w_0\|^2)$.

Then the interpolated SDF $f(z; \hat{w}_\alpha)$ approximates the signed distance function of the mesh defined by the control points $C(\hat{p}_\alpha)$, where $\hat{p}_\alpha = \sum_i \alpha_i p_i$. That is,

$$f(z; \hat{w}_\alpha) \approx \text{SDF}(z; C(\hat{p}_\alpha)).$$

**Proof.** By assumption (2), for each $i$,

$$f(z; w_i) \approx d_i(z) = \text{SDF}(z; C(p_i)).$$

Then by local linearity of $f$ in weights (3),

$$f(z; \hat{w}_\alpha) \approx \sum_i \alpha_i f(z; w_i) \approx \sum_i \alpha_i d_i(z).$$

Now, because control points interpolate linearly by assumption (1), we define:

$$C_\alpha := \sum_i \alpha_i C(p_i) = C\left(\sum_i \alpha_i p_i\right) = C(\hat{p}_\alpha).$$

If the SDFs $d_i(z)$ correspond to shapes with shared topology and smooth variation in geometry, then the signed distances satisfy:

$$\sum_i \alpha_i d_i(z) \approx \text{SDF}(z; C_\alpha).$$

Therefore,

$$f(z; \hat{w}_\alpha) \approx \text{SDF}(z; C(\hat{p}_\alpha)).$$

Thus, the zero-level set of $f(z; \hat{w}_\alpha)$ corresponds to the mesh defined by $\hat{p}_\alpha$, completing the proof.

## C APPROXIMATE LINEARITY OF THE SDF DECODER IN WEIGHTS

Let $z \in \mathbb{R}^3$ be a fixed 3D input point, and let $\gamma(z) \in \mathbb{R}^D$ denote its Fourier positional encoding, defined as:

$$\gamma(z) = \left[z, \sin(2^0 \pi z), \cos(2^0 \pi z), \ldots, \sin(2^L \pi z), \cos(2^L \pi z)\right].$$

Let $f(z; w)$ be a feedforward multilayer perceptron (MLP) with parameters $w$ and input $\gamma(z)$. The network is composed of $K$ layers with weights and biases $\{W_k, b_k\}_{k=1}^K$, where:

$$\begin{aligned}
h_0 &= \gamma(z), \\
h_k &= \phi(W_k h_{k-1} + b_k), \quad \text{for } k = 1, \ldots, K-1, \\
f(z; w) &= W_K h_{K-1} + b_K,
\end{aligned}$$

with $\phi(\cdot)$ a fixed elementwise nonlinearity (e.g., ReLU). The parameter vector $w$ collects all $\{W_k, b_k\}$.

**Claim.** For fixed $z$, if all weights $\{w_i\}_{i=1}^N$ lie in a small neighborhood of a reference $w_0$, then $f(z; w)$ is approximately linear in $w$. In particular, for convex coefficients $\{\alpha_i\}_{i=1}^N$ with $\sum_{i=1}^N \alpha_i = 1$, we have

$$f\left(z; \sum_{i=1}^N \alpha_i w_i\right) \approx \sum_{i=1}^N \alpha_i f(z; w_i),$$

with an error term of order $O(\|w_i - w_0\|^2)$.

**Proof.** Fix the input $z$, so $\gamma(z)$ is constant. Consider the Taylor expansion of $f(z; w)$ about $w_0$:

$$f(z; w) = f(z; w_0) + \nabla_w f(z; w_0)^\top (w - w_0) + R(w),$$

where $R(w)$ is the second-order remainder term.

Applying this to each $w_i$ gives

$$f(z; w_i) = f(z; w_0) + \nabla_w f(z; w_0)^\top (w_i - w_0) + R(w_i).$$

Now evaluate at the convex combination $\hat{w}_\alpha = \sum_i \alpha_i w_i$:

$$f(z; \hat{w}_\alpha) = f(z; w_0) + \nabla_w f(z; w_0)^\top \left(\sum_i \alpha_i (w_i - w_0)\right) + R(\hat{w}_\alpha).$$

On the other hand, the convex combination of outputs is

$$\sum_i \alpha_i f(z; w_i) = f(z; w_0) + \nabla_w f(z; w_0)^\top \left( \sum_i \alpha_i (w_i - w_0) \right) + \sum_i \alpha_i R(w_i).$$

Subtracting the two expressions gives

$$f(z; \hat{w}_\alpha) - \sum_i \alpha_i f(z; w_i) = R(\hat{w}_\alpha) - \sum_i \alpha_i R(w_i).$$

Since the remainder terms $R(\cdot)$ are second-order in the deviations $(w_i - w_0)$, this difference is $O(\max_i \|w_i - w_0\|^2)$. Thus, when all weights are close, the error is small and the decoder behaves approximately linearly in $w$.

# D  ADDITIONAL QUANTITATIVE AND QUALITATIVE RESULTS

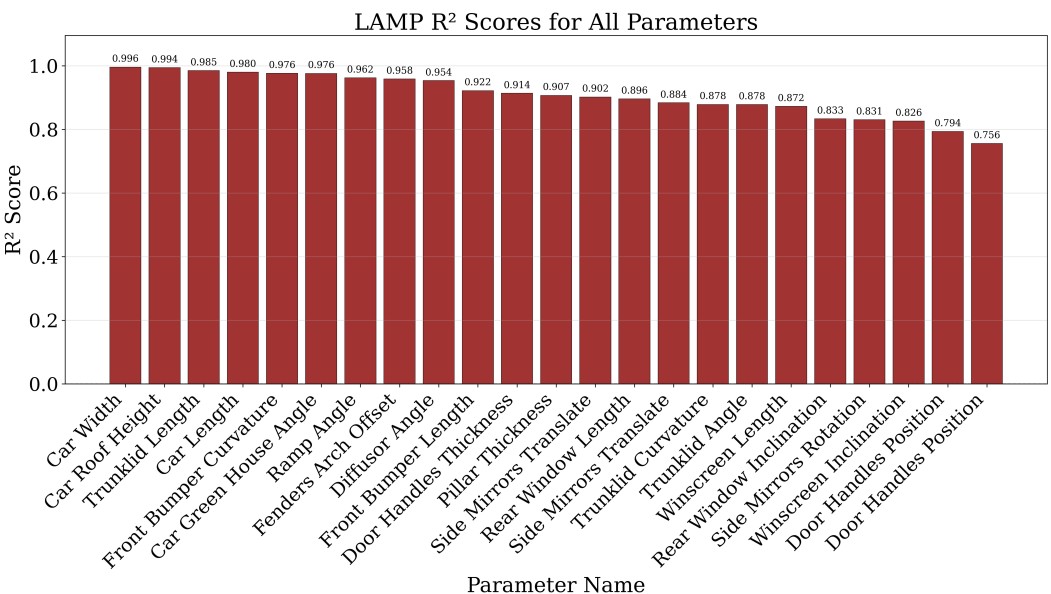

Figure 7: LAMP's $R^2$ scores for single-parameter sweeps on the DrivAerNet++ dataset, extrapolated $\pm 100\%$ beyond the dataset range.

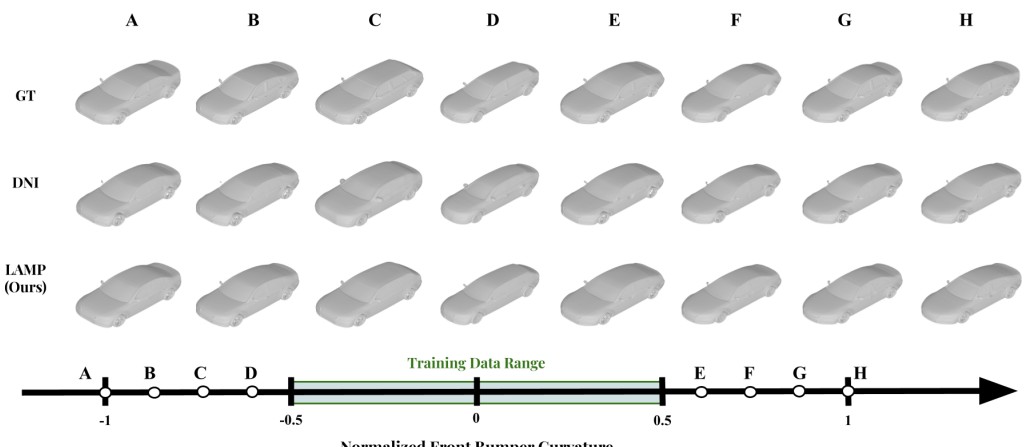

Figure 8: Single-parameter extrapolation within dataset range on DrivAerNet++. Training samples are restricted to a centered 50% interval of the parameter, while evaluation is performed outside this interval. LAMP maintains smooth, plausible extrapolation, while DNI drifts away from target shapes.

Table 5: Interpolation performance. We compare LAMP against DNI and AE-LPA baselines on BlendedNet using Chamfer Distance (CD ↓), Intersection-over-Union (IoU ↑, in %), and surrogate-based parameter error (↓). Training uses 100 samples from the dataset, and testing uses 200 held-out samples.

| Dataset | Method | # Samples | CD ↓ | IoU ↑ (%) | MAE ↓ |
|---|---|---|---|---|---|
| | DNI | 100 | 0.0346 | 94.21 | 0.0038 |
| **BlendedNet** | AE-LPA | 100 | 0.0393 | 88.26 | 0.0078 |
| | **LAMP (Ours)** | 100 | **0.0172** | **95.35** | **0.0031** |

Table 6: Large-range extrapolation (BlendedNet) up to $\pm 50\%$. LAMP sustains high fidelity ($R^2 > 0.78$) for both single- and multi-parameter extrapolation. Training uses 100 samples from the dataset. For single-parameter extrapolation, we sample 10 values uniformly per parameter within the extrapolated range, constraining that parameter while allowing the others to vary. For multi-parameter extrapolation, we repeat 100 trials where four parameters are randomly selected and set outside the dataset range (up to $50\%$ extrapolation, i.e., twice the original span).

| Dataset | Method | Single Parameter | | | Multi-Parameter | | |
|---|---|---|---|---|---|---|---|
| | | MMD $\downarrow$ | MAE $\downarrow$ | $R^2 \uparrow$ | MMD $\downarrow$ | MAE $\downarrow$ | $R^2 \uparrow$ |
| **BlendedNet** | DNI | 0.038 | 0.392 | 0.784 | 0.040 | 0.435 | 0.521 |
| | AE-LPA (100) | 0.039 | 0.611 | 0.169 | 0.043 | 0.823 | -0.069 |
| | **LAMP (Ours)** | **0.035** | **0.305** | **0.868** | **0.037** | **0.353** | **0.782** |

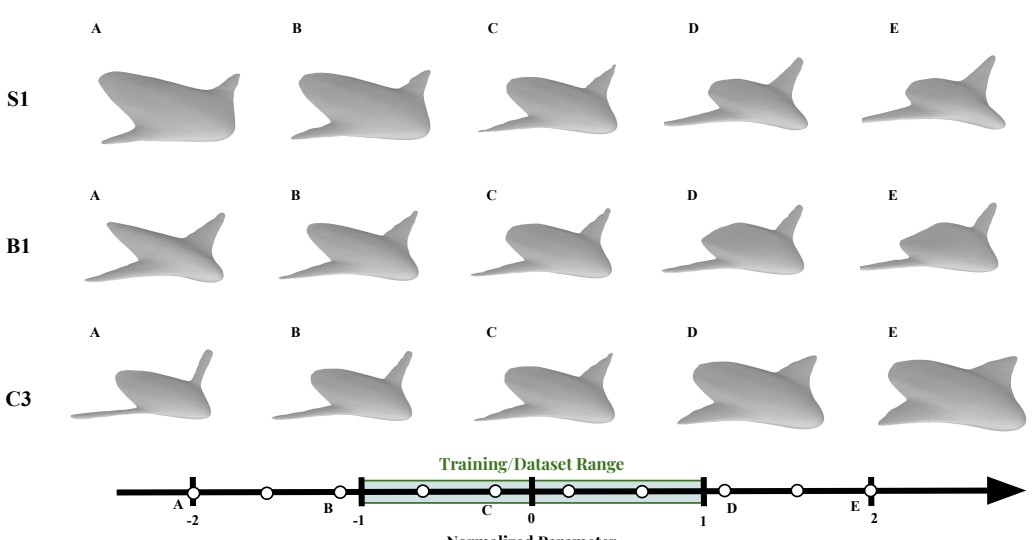

Figure 9: Single-parameter sweep on BlendedNet. LAMP sustains smooth, plausible geometry under large parameter shifts.

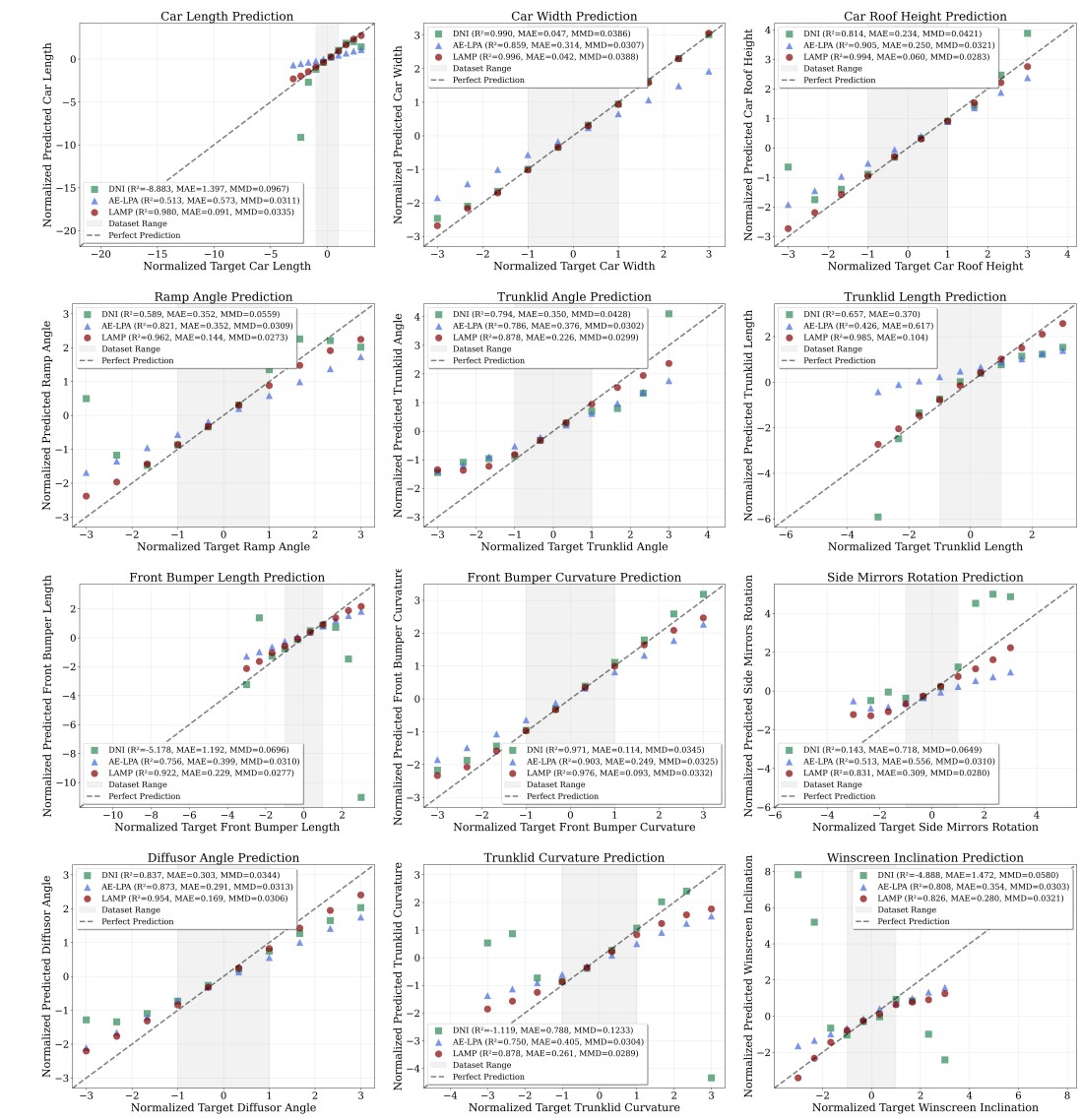

Figure 10: Single-parameter extrapolation beyond the dataset range. All other parameters are allowed to vary. The plots show surrogate-predicted versus target parameters, comparing LAMP against the baselines DNI and AE-LPA.

# E ABLATION STUDY: HOW DOES SAMPLE SIZE AFFECT RELIABILITY AND EXTRAPOLATION IN LAMP?

We ablate the effect of sample size in Table 7. As the number of samples increases, MAE decreases while both $R^2$ and the mean safe extrapolation range (%) increase, before plateauing at larger sample counts. This trend indicates that performance improves with more samples but saturates beyond a certain scale.

Table 7: Ablation study on mixing quality and reliability across different numbers of samples using LAMP

| Number of Samples | $R^2$ ↑ | MAE ↓ | Mean Safe Extrapolation Range (%) ↑ |
|---|---|---|---|
| 10 | -7.289 | 2.650 | 145.8 |
| 50 | -0.214 | 1.083 | 213.9 |
| 100 | 0.838 | 0.507 | 330.6 |
| 500 | 0.849 | 0.479 | 418.1 |
| 1000 | 0.862 | 0.486 | 427.8 |

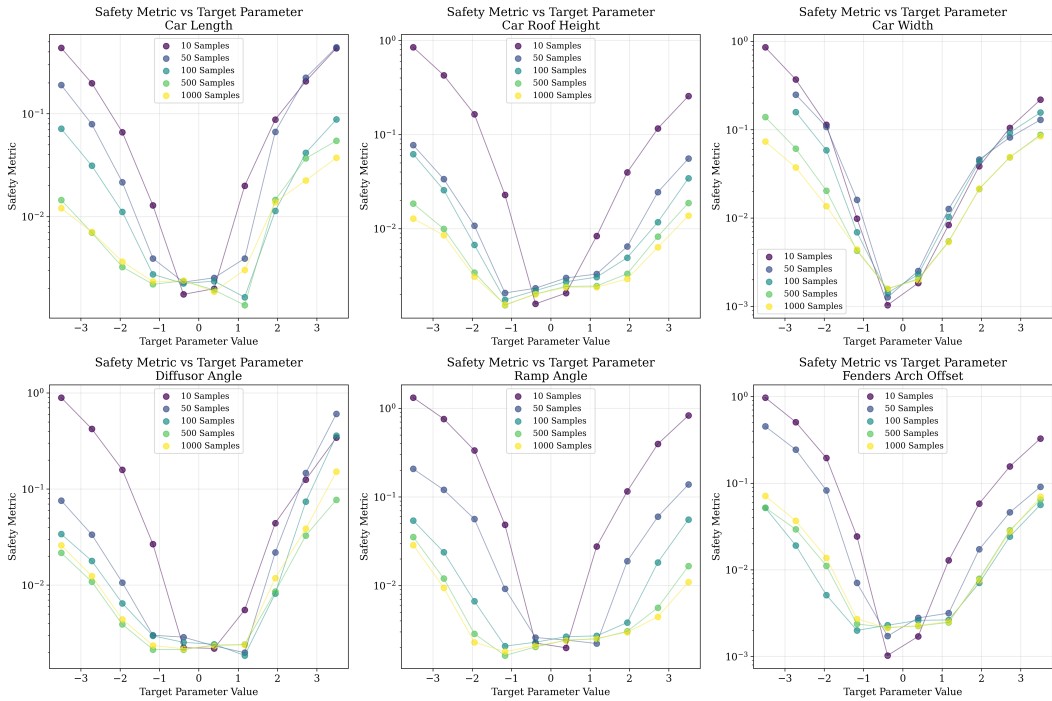

Figure 11: Safety metric values as a function of target parameter sweeps across six design parameters in DrivAerNet++. Curves correspond to different training set sizes (10–1000 samples). Larger datasets consistently reduce the safety metric, indicating more reliable extrapolation across parameter ranges.

## F    ABLATION STUDY: HOW DOES SDF DECODER FINETUNING AFFECT RELIABILITY AND EXTRAPOLATION IN LAMP?

### SENSITIVITY TO INITIALIZATION

To evaluate whether LAMP depends on a shared initialization during SDF overfitting, we trained exemplar SDF decoders both with and without a common initialization. With a shared initialization, all exemplars remain in a common local basin, preserving strong weight-space alignment and enabling smooth, parameter-consistent interpolations.

Removing the shared initialization forces each decoder to converge to a different basin, breaking weight-space alignment. Under this setting, affine interpolation produces severely distorted and non–car-like shapes, confirming that a shared initialization is essential for maintaining a coherent basis.

### ROBUSTNESS TO RANDOM SEEDS

We next tested whether varying the random seed (while keeping the shared initialization fixed) affects weight-space alignment. Using different seeds slightly reduces performance but does not break alignment: generated shapes remain visually consistent, parameter predictions stay stable, and the linearity assumption (A2) continues to hold.

In contrast, removing the shared initialization destroys linearity entirely and leads to invalid, severely distorted generations.

To quantify reliability, we measured $R^2$, MAE, and safe extrapolation range over single-parameter sweeps using 100 exemplars. Results are summarized in Table 8.

Table 8: Effect of initialization and random seeds on reliability and extrapolation behavior in LAMP, evaluated over 100 exemplar SDF decoders.

| Condition | $R^2$ | MAE | Mean Safe Extrapolation Range |
|---|---|---|---|
| Same Seed | 0.838 | 0.507 | 330% |
| Random Seed | 0.773 | 0.610 | 304% |
| No Initialization | −37.77 | 9.16 | 0% |

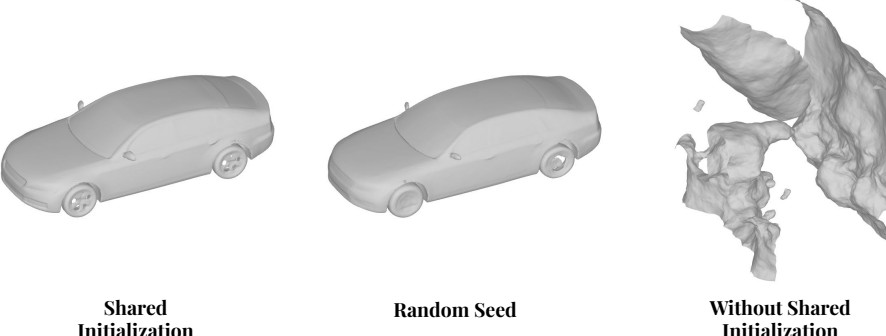

| **Shared Initialization** | **Random Seed** | **Without Shared Initialization** |

Figure 12: Representative comparison of weight-space interpolation under different training conditions: **Left:** shared initialization (stable, coherent interpolation). **Middle:** random seeds (still stable). **Right:** no shared initialization (interpolation collapses).

## G    EVALUATION METRICS

- **Chamfer Distance (CD):**

$$\text{CD}(X,Y) = \frac{1}{|X|} \sum_{x \in X} \min_{y \in Y} \|x - y\|_2^2 \; + \; \frac{1}{|Y|} \sum_{y \in Y} \min_{x \in X} \|y - x\|_2^2,$$

where $X$ and $Y$ are point clouds sampled from the predicted and reference meshes.
- **Intersection-over-Union (IoU):**

$$\text{IoU}(A,B) = \frac{|A \cap B|}{|A \cup B|},$$

where $A$ and $B$ are voxelizations of the predicted and reference meshes.
- **Minimum Matching Distance (MMD):**

$$\text{MMD}(S_g, S_r) = \frac{1}{|S_g|} \sum_{x \in S_g} \min_{y \in S_r} d(x,y),$$

where $S_g$ and $S_r$ are sets of generated and reference point clouds, and $d(\cdot, \cdot)$ is typically the Chamfer distance between individual shapes.

## H    CONSTRAINT COMPLIANCE VALIDATION: SURROGATES AND DIRECT MEASUREMENTS

**Comparison of Mesh-Based Surrogates.**    In the main text, we validated design-parameter compliance using a mesh-based surrogate model: we fixed random PointNet embeddings of each decoded mesh (deterministic initialization) and fit a LASSO regressor to predict physical parameters. Interestingly, this simple surrogate achieves strong accuracy ($R^2 > 0.9$ on held-out test sets), despite the encoder being untrained.

To test whether stronger pretrained representations improve performance, we compared against the OpenShape point cloud embedding model Liu et al. (2023). Across all parameters, the OpenShape-based surrogate achieved consistently lower $R^2$ scores than the randomly initialized PointNet embeddings. This suggests that domain-specific geometric structure is better captured by lightweight randomized encoders than by pretrained embeddings trained on natural 3D categories.

Figure 13 illustrates the surrogate pipeline. Figures 15 and 14 show predicted versus ground-truth parameter values on the BlendedNet and DrivAerNet++ datasets, respectively, demonstrating high accuracy across test sets. We train the surrogates on 800 samples and evaluate on a held-out test set of 200 samples.

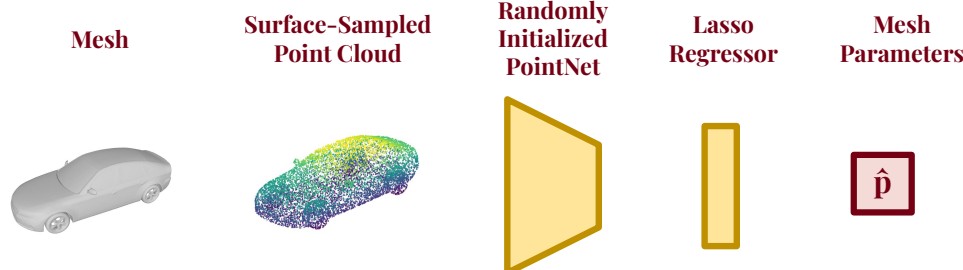

Figure 13: Diagram of the mesh-based surrogate pipeline. A decoded mesh is first converted into a surface-sampled point cloud. The point cloud is passed through a randomly initialized PointNet encoder to produce fixed embeddings, which are then mapped to interpretable mesh parameters via a LASSO regressor. This simple pipeline achieves strong predictive accuracy ($R^2 > 0.9$) despite the encoder being untrained.

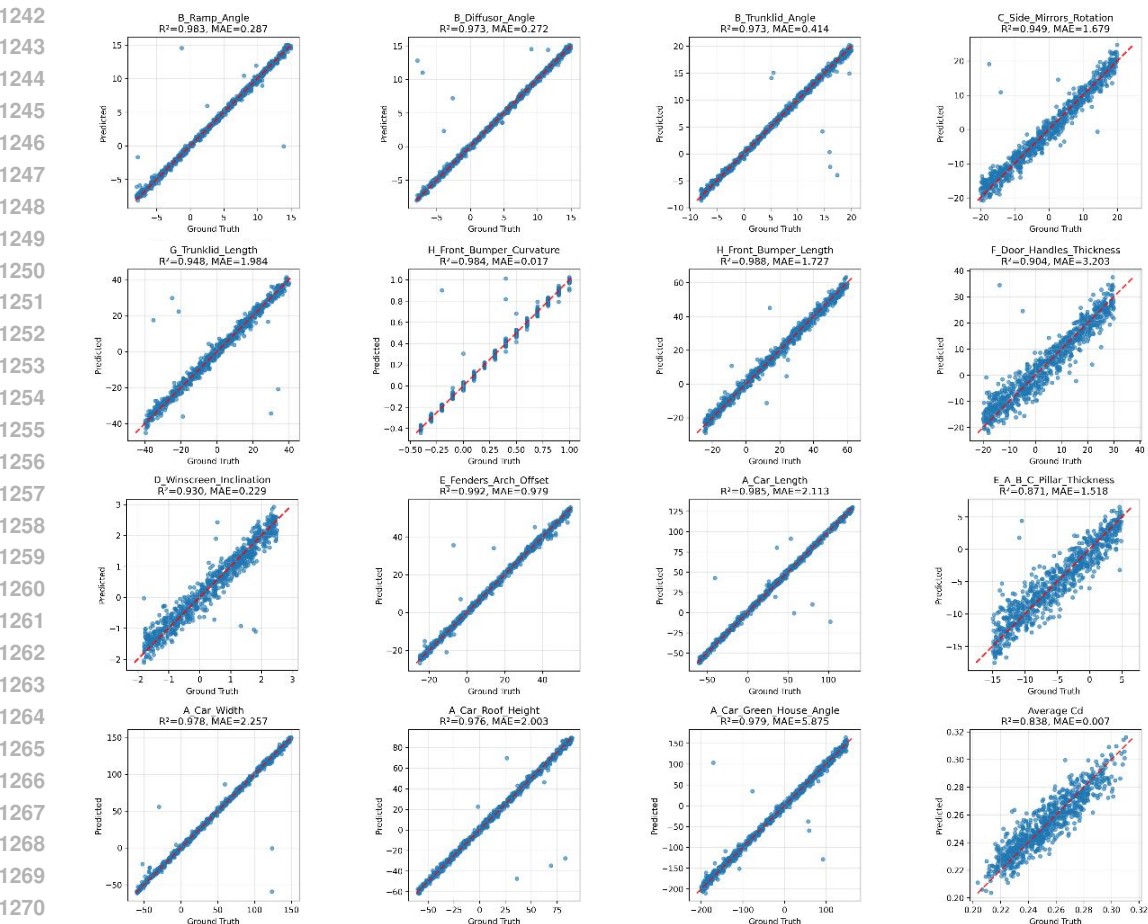

Figure 14: Predicted vs. ground truth parameters on the DrivAerNet++ test set, evaluating the mesh-based surrogate model for parameter prediction.

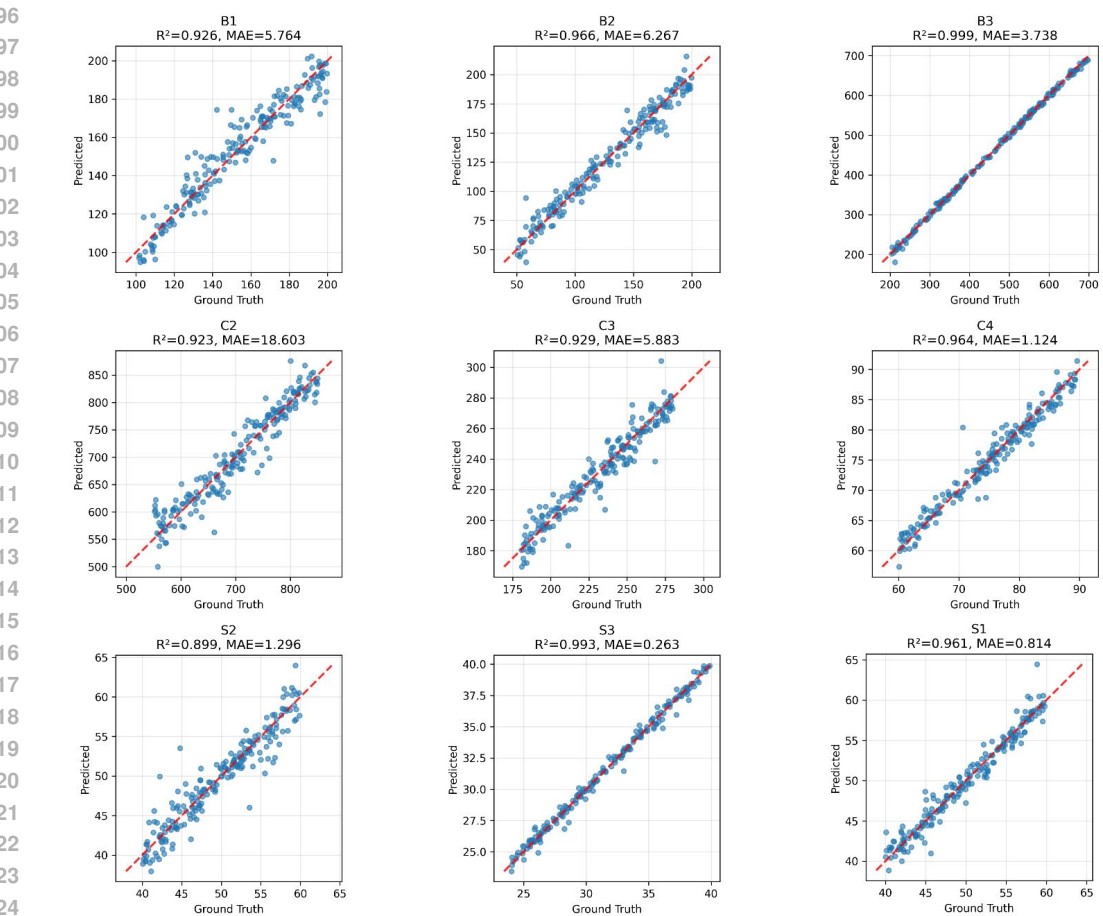

Figure 15: Predicted vs. ground truth parameters on the BlendedNet test set, evaluating the mesh-based surrogate model for parameter prediction.

**Direct Geometric Measurements.** Beyond surrogate-based validation, we also implemented direct geometric measurements for certain parameters. For example, to compute *Car Length* on DrivAerNet++ cars, we measure the distance between the centers of the front and rear wheels. Specifically:

1. We take slices of the decoded SDF along the wheel plane.
2. We detect circular cross-sections with radii in the expected range of wheel radii.
3. We identify the front and rear wheel centers and compute their distance.
4. We map this distance back to the labeled Car Length value by calibrating on ground-truth SDFs from the dataset.

This method provides a parameter-compliant, geometry-based validation of mesh outputs. Figure 16 shows an example of wheel detection and length estimation on generated meshes.

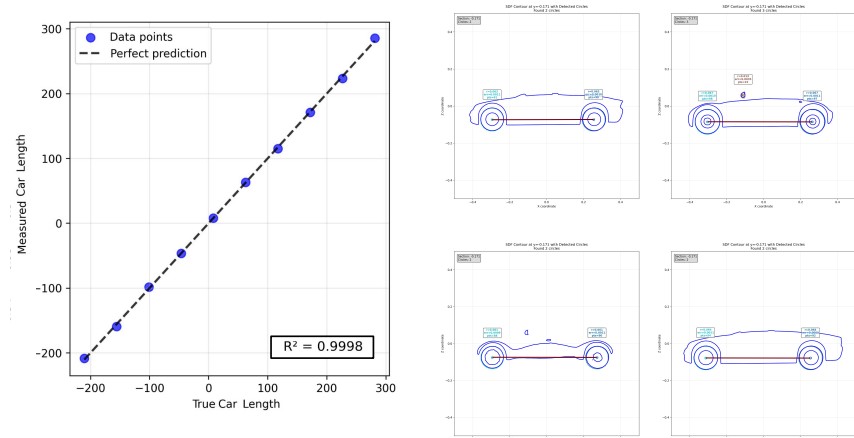

Figure 16: Direct geometric measurement of car length in DrivAerNet++. We slice the decoded SDF, detect wheel cross-sections by circle fitting, compute the distance between wheel centers, and map this measurement back to the dataset-defined Car Length parameter.

# I   VALIDATION OF THE LINEARITY-MISMATCH METRIC AGAINST HUMAN ANNOTATED DATA

**Assumption.** Our safety metric relies on the assumption that the decoder $f(z; w)$ is locally linear in weights $w$ for a fixed spatial coordinate $z$. Formally,

$$f\left(z; \sum_{i=1}^{N} \alpha_i w_i\right) \approx \sum_{i=1}^{N} \alpha_i f(z; w_i),$$

with approximation error bounded by $O\big(\max_i \|w_i - w_0\|^2\big)$. This implies that as long as interpolations in weight space remain sufficiently close to the training exemplars, affine mixing should yield faithful mesh reconstructions. The linearity mismatch defined in Sec. 3 measures deviations from this assumption.

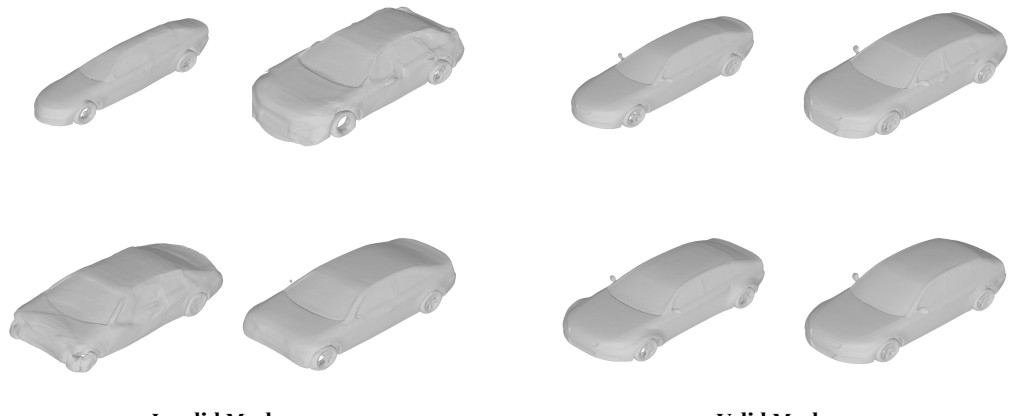

**Invalid Meshes**                              **Valid Meshes**

Figure 17: Examples of meshes labeled during human annotation. Left: *Invalid meshes*, which exhibit collapsed, distorted, or implausible geometries. Right: *Valid meshes*, which maintain smooth, realistic car shapes with high geometric fidelity. These labels are used as ground truth to validate the safety metric.

**Dataset Construction.** To empirically validate this assumption, we constructed a diagnostic dataset by systematically varying one shape parameter at a time. Each parameter was interpolated and extrapolated up to a $700\%$ ($\pm 300\%$) increase in range compared to its span in the main dataset. For every setting, we decoded a mesh $\mathcal{M}_d$ using mixed weights $\mathbf{w}_d = \sum_i \alpha_i w_i$ and computed the linearity-mismatch score. (Fig. 17)

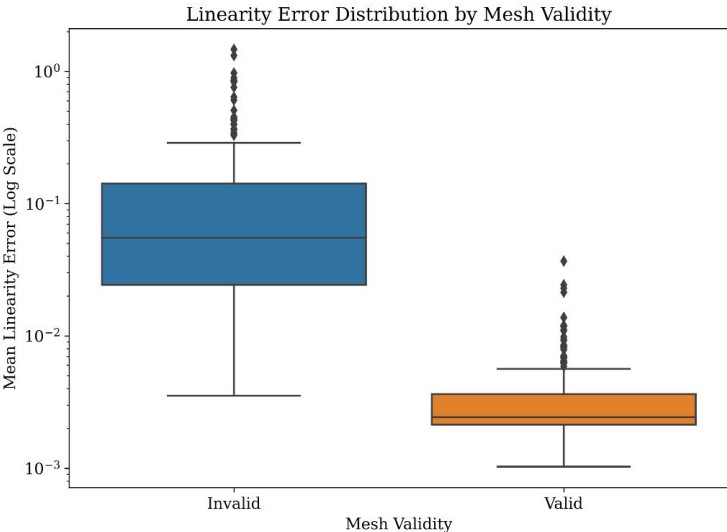

Figure 18: Box plot of mean linearity error (log scale) across meshes labeled as valid vs. invalid. Valid meshes concentrate at low mismatch values, while invalid meshes show significantly higher errors, confirming that the linearity-mismatch metric is a strong predictor of mesh validity.

**Human Annotation Protocol.** All meshes were visually inspected and annotated as either *valid* or *invalid*. A mesh was considered valid if it was smooth and resembled a high-fidelity car geometry without collapse or severe distortion. Invalid meshes were those with degenerate or implausible deformations. This produced a binary ground-truth dataset for evaluation.

**Metric Validation.** We used the mismatch score to predict mesh validity and compared it against human annotations:

- The ROC curve (Fig. 19, top left) shows excellent discriminative power with an area under the curve (AUC) of **0.989**.
- The precision–recall curve (Fig. 19, top right) yields an AUC of **0.990**, indicating reliable separation of valid from invalid meshes.
- Threshold analysis (Fig. 19, bottom) reveals that $\epsilon = 0.01$ provides a good tradeoff, achieving high recall while preserving precision.

**Distributional Analysis.** To further assess robustness, we examined the distribution of linearity errors across mesh validity labels. As shown in Fig. 18, valid meshes cluster at low mismatch values, while invalid meshes exhibit substantially higher errors, confirming that the safety metric is well aligned with perceptual mesh quality.

**Discussion.** These experiments demonstrate that the linearity-mismatch metric is a reliable quantitative proxy for mesh validity. Its strong agreement with human annotations justifies our use of $\epsilon = 0.01$ as the default safety threshold throughout this work. Moreover, the threshold can be strict or relaxed depending on the application's tolerance for geometric distortion. In aerodynamic design studies or performance-critical evaluations, we adopt a conservative cutoff to avoid any degradation in shape fidelity. In contrast, exploratory design settings may permit larger mismatch values to encourage diversity in the generated geometries. Practitioners may therefore choose a threshold smaller or larger than 0.01 depending on the demands of their downstream task.

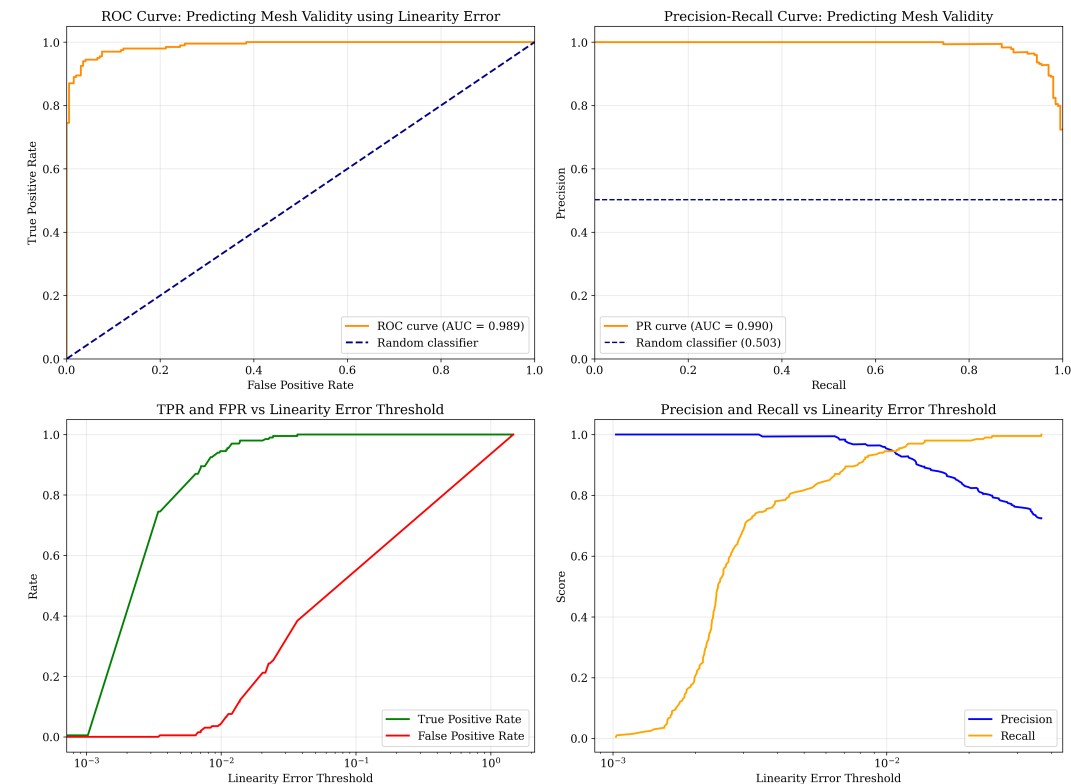

Figure 19: Validation of the linearity-mismatch safety metric against human-annotated mesh validity. Top left: ROC curve showing high discriminative power (AUC = 0.989). Top right: Precision–recall curve (AUC = 0.990). Bottom left: true positive rate (TPR) and false positive rate (FPR) as a function of linearity error threshold. Bottom right: precision and recall as a function of threshold. Together, these results confirm that the safety metric reliably predicts mesh validity, with $\epsilon = 0.01$ providing a good tradeoff between precision and recall.

## J  PARAMETRIZATION OF DRIVAERNET++ AND BLENDEDNET

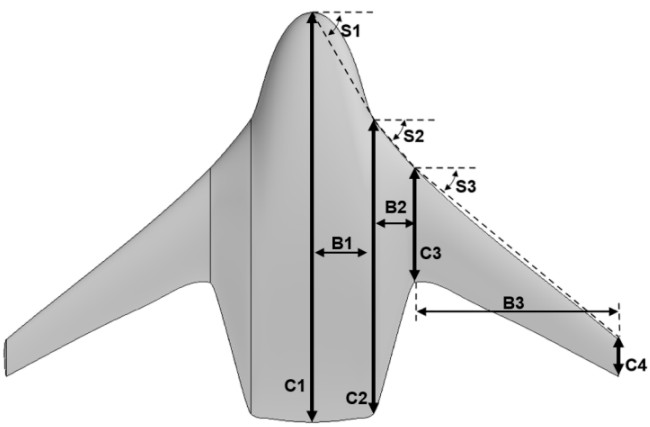

Figure 20: BlendedNet Parametrization Sung et al. (2025)

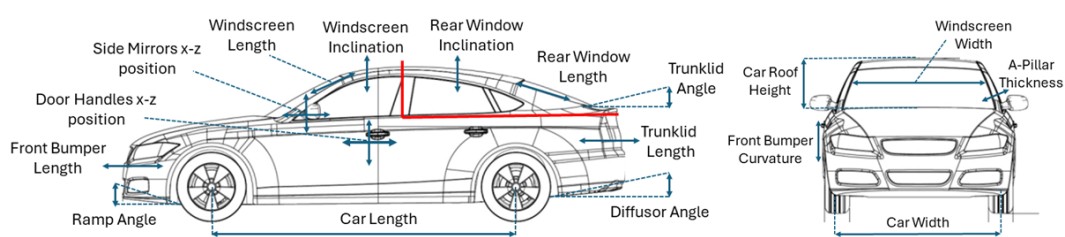

Figure 21: DrivAerNet++ Parametrization Elrefaie et al. (2024)

# K  QUANTITATIVE VALIDATION OF LINEARITY ASSUMPTION (A2)

To assess the linearity assumption (A2), we compared the signed distance field (SDF) predicted by the interpolated network,

$$f\left(z; \sum_i \alpha_i w_i\right),$$

against the linear combination of individual model outputs,

$$\sum_i \alpha_i f(z; w_i).$$

We sampled 10,000 random 3D query points and evaluated both quantities across a range of interpolation and mild extrapolation factors. As shown in Fig. 22, the relationship is nearly perfectly linear in-distribution, with $R^2 \approx 0.99$. Linearity degrades gradually as the extrapolation factor increases.

This empirical analysis demonstrates that SDF networks exhibit approximate linear behavior under weight-space interpolation, supporting assumption (A2).

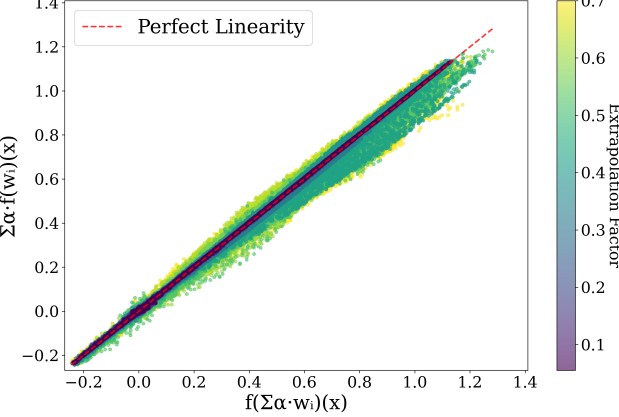

Figure 22: Linearity comparison of $f(z; \sum_i \alpha_i w_i)$ vs. $\sum_i \alpha_i f(z; w_i)$ evaluated over 10,000 3D query points while varying the extrapolation factor.

## L EXTENDING LAMP TO NON-IMPLICIT DECODERS

To examine whether LAMP can be applied beyond implicit SDF models, we performed an experiment using a non-implicit point cloud decoder that operates by deforming a spherical template into the target point cloud. First, we overfit this deformation-based decoder to a single target shape (left). We then fine-tuned the same model on a second target shape (middle), producing two distinct sets of network weights corresponding to two different geometries.

With these two trained decoders, we linearly interpolated between their weight vectors and decoded the resulting intermediate geometry. As shown in Fig. 23, the interpolated weights produce a smooth and coherent shape that lies between the two endpoints (right), confirming that weight-space interpolation remains valid even in non-implicit architectures.

This experiment demonstrates that the LAMP principle, leveraging linearity and smoothness under weight interpolation for controllable geometry manipulation, extends naturally to deformation-based point cloud decoders. This suggests that LAMP provides a general mechanism for parameterized geometry control across diverse neural shape representations.

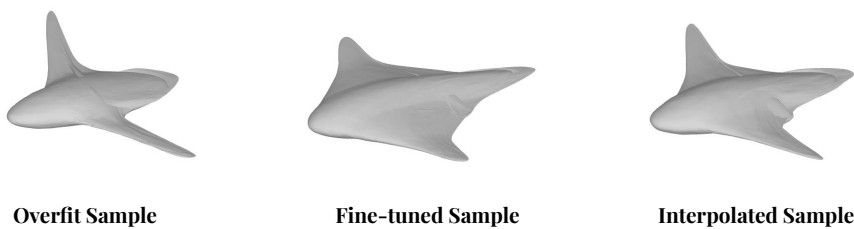

**Overfit Sample**          **Fine-tuned Sample**          **Interpolated Sample**

Figure 23: Interpolation using a deformation-based point cloud decoder. **Left:** model overfit to the first target (sphere deformed into shape A). **Middle:** model fine-tuned on the second target (sphere deformed into shape B). **Right:** linear interpolation in weight space yields a valid intermediate geometry, showing that LAMP extends beyond implicit SDF decoders.

# M   MULTI-OBJECTIVE LAMP: BALANCING PARAMETER SATISFACTION AND SAFE GENERATION

### FORMULATION

A natural extension of LAMP is to treat parameter matching and geometric reliability as a multi-objective optimization. Instead of minimizing only the parameter error, we solve

$$\min_{\alpha} \left\| \mathbf{P}_{;,\mathcal{C}}^{\top}\alpha - \mathbf{p}_{d,\mathcal{C}} \right\|_2^2 \; + \; \lambda \left\| \alpha \right\|_2^2 \quad \text{s.t.} \quad \mathbf{1}^{\top}\alpha = 1. \tag{3}$$

where $\|\alpha\|_2$ serves as a fast extrapolation proxy. Evaluating the full linearity-mismatch safety metric at each iteration requires evaluating the $N$ unflattened SDF decoders for each of the 3D query points, which is computationally heavy. In contrast, $\|\alpha\|$ varies monotonically with extrapolation and correlates strongly with the final safety score, enabling efficient exploration of accuracy vs reliability trade-offs. Since LAMP's optimization is extremely fast, sweeping $\lambda$ yields the Pareto frontier essentially for free, letting users select conservative, balanced, or exploratory solutions depending on application tolerance.

### CONFLICTING PARAMETERS AND SAFETY BEHAVIOR

Conflicting or mutually incompatible parameter targets often force the optimization outside the valid linear regime, producing large $\|\alpha\|$ and correspondingly high safety scores. The leftmost example in Fig. 24 shows such a case: although the parameter error (MAE) is low, the required extrapolation produces visible geometric distortion. When decoded into meshes, these settings exhibit collapse or deformation, and the linearity-mismatch metric correctly flags them as invalid. This multi-objective formulation provides a principled way to avoid such failure modes while still permitting controlled extrapolation for creative or exploratory design tasks.

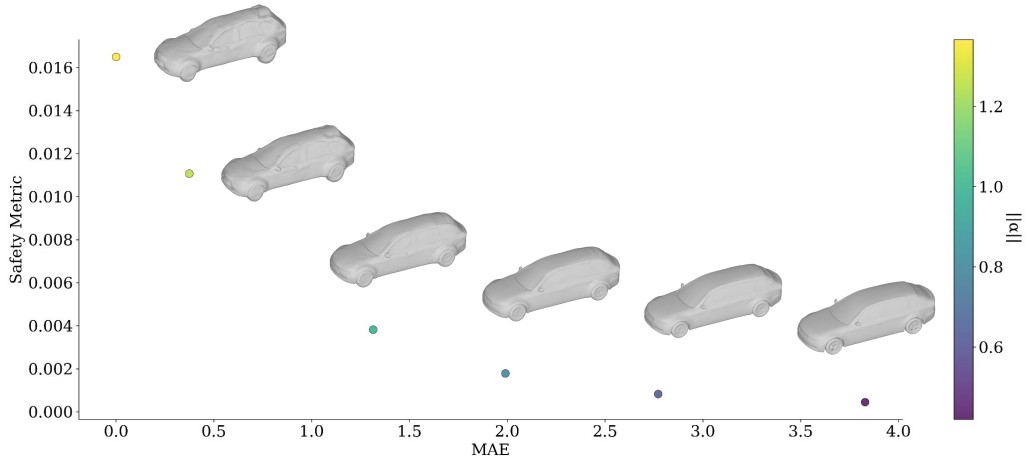

Figure 24: Trade-off between parameter error (MAE), extrapolation ($\|\alpha\|$), and geometric safety. The leftmost point illustrates conflicting parameter constraints: despite a low MAE, the high $\|\alpha\|$ and safety metric indicate that the solution lies outside the valid linear region, leading to distorted geometries after decoding.

# N    QUANTIFYING LAMP'S HIGH-DIMENSIONAL EXTRAPOLATION VOLUME

To quantify how LAMP expands the reachable design space under controlled extrapolation, we perform the analysis on a subset of 23 parameters. Each selected parameter is scaled to lie in $[-1, 1]$, and extrapolation corresponds to enlarging this 23-dimensional hypercube to

$$[-1 - f, \ 1 + f]^{23},$$

which has volume $(1 + f)^{23}$ relative to the original domain.

### MONTE CARLO APPROXIMATION

For each extrapolation factor $f$, we uniformly sample $10{,}000$ random parameter vectors in the expanded domain $[-1 - f, 1 + f]^{23}$. For each sample, we solve for mixture coefficients using LAMP and generate the corresponding mesh. We then evaluate the linearity-mismatch safety metric to determine whether the geometry is valid.

Let Valid($f$) denote the fraction of samples whose meshes pass the safety threshold. The **extrapolation volume ratio** for factor $f$ is then estimated as

$$\text{VolRatio}(f) \ = \ (1 + f)^{23} \times \text{Valid}(f),$$

providing an aggregate measure of how much of the enlarged parameter space remains safely reachable.

### RESULTS

LAMP provides reliable parameter control even under extrapolation. Using only 100 normalized exemplar shapes, LAMP already achieves high-accuracy mixing and maintains strong local linearity. As shown in Fig. 25b, extrapolating each parameter by just 50% expands the reachable design space by a factor of $\sim 8000\times$ while still preserving a high validity ratio under the safety metric. This demonstrates that modest extrapolation combined with sufficient coverage of the parameter space enables both accurate control and large-scale geometry exploration.

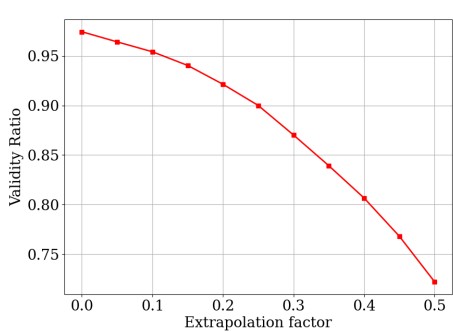 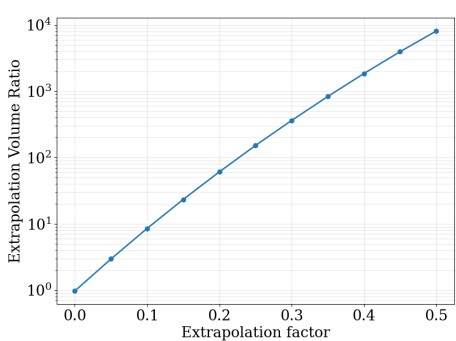

(a) Validity ratio vs. extrapolation factor.          (b) Extrapolation volume ratio.

Figure 25:    (a) Validity ratio under increasing extrapolation factor $f$.    (b) Corresponding extrapolation-volume ratio in the 23-D normalized parameter space. LAMP achieves accurate control with only 100 samples and, with a 50% extrapolation ($f = 0.5$), enables exploration of a design space approximately **8000× larger** than the dataset volume.

## O   MIXING BEHAVIOR ON GEOMETRIES WITH TOPOLOGICAL FEATURES

To assess whether LAMP can reliably handle shapes that include topological features such as holes, we conducted a controlled study on a simple but representative part: a rectangular plate with a circular through-hole. The geometry is parameterized by six parameters: plate *length*, *width*, and *thickness*, and hole *radius*, and hole *x*- and *y*-positions.

We trained a set of 13 exemplar SDF decoders from a shared initialization and applied LAMP to generate new shapes by varying each parameter independently. For every parameter, we performed a sweep that extends beyond the training range by $\pm100\%$ of the original dataset bounds.

As shown in Fig. 26, LAMP successfully produces valid, smoothly varying geometries throughout these extended sweeps. In particular, the hole feature remains stable under extrapolation, and its radius and position vary as expected even when moved well outside the dataset range. These results demonstrate that LAMP is capable of mixing weights for shapes that contain localized geometric and topological features, maintaining structural coherence even under aggressive extrapolation.

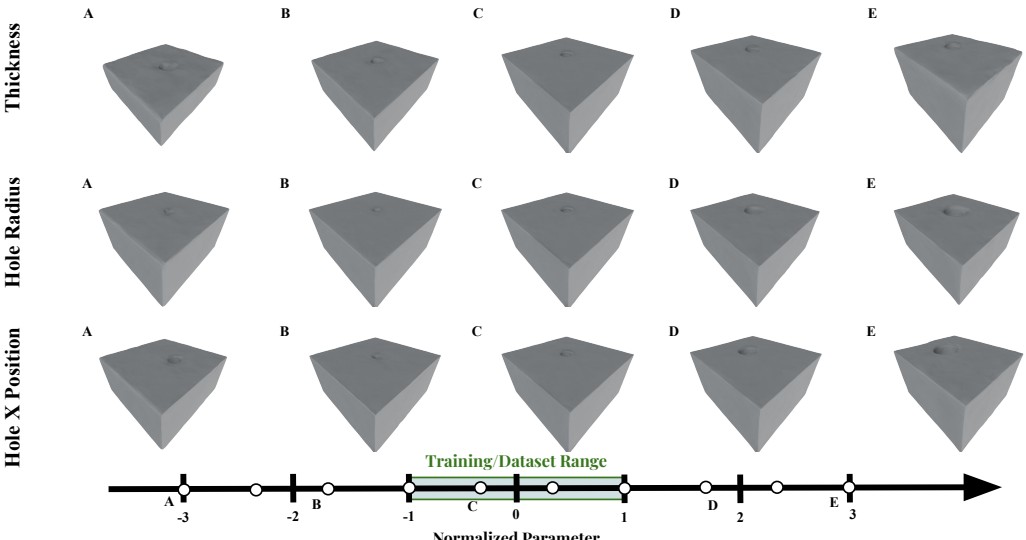

Figure 26: Parameter sweeps on a plate-with-hole geometry using LAMP. Each column (A–E) corresponds to normalized parameter values extending to $\pm100\%$ beyond the dataset range. LAMP maintains coherent geometry and stable topological structure across all extrapolations.

## P   LLM USAGE

Large Language Models (LLMs) were used to aid or polish writing. The authors reviewed and take full responsibility for the content.

