# OpenReview forum: "LAMP: Data-Efficient Linear Affine Weight-Space Models for Parameter-Controlled 3D Shape Generation and Extrapolation"
_ICLR.cc/2026/Conference — ICLR 2026 Conference Desk Rejected Submission_

### Official Review · Reviewer_CbJm · 2025-10-23

**Soundness:** 3
**Presentation:** 3
**Contribution:** 3
**Rating:** 8
**Confidence:** 3

**Summary:**

This paper proposes a 3D shape generation method that aims to produce shapes that respect certain parameter constraints.
This is implemented by computing an affine combination of parameter vectors of known shapes under the given constraints.
The shape is then generated by using the same weights to combine the weights of neural SDF representations of the corresponding known shapes.
To check the validity of the generated shape, a linearity-mismatch metric is proposed.

Thus, the contribution of this paper is a new method for controlled interpolation and extrapolation of a limited set of shapes.
Experiments show that the proposed method can generate shapes that respect the given constraints, and that the linearity-mismatch metric correlates well with the quality of the generated shapes.

**Strengths:**

The strength of this paper lies in the novel idea of combining parameter vectors and neural SDF weights to generate new shapes under given constraints.

The method offers some much-needed fine-grained control over the generated shapes.
Furthermore, the proposed method does not rely on large datasets or compute for training.

The method is conceptually simple and easy to implement, and the experiments demonstrate that the affine combination weights derived from the parameter vectors can (perhaps somewhat surprisingly) be effectively used to generate new shapes that respect the constraints.

Overall the paper is well-written and clearly presents the proposed method and experimental results.

**Weaknesses:**

One weakness of the paper is that the linearity of the SDF weights is only reasonable for very similar weights.
This limits the variability of the generated shapes.

Furthermore, it is not clear to me how sensitive the method is to the choice and number of known shapes.
If the known shapes do not cover the parameter space well, it is unclear how well the method performs.

On a more minor note Table 1 does not seem to be referenced in the text.

**Questions:**

* How are the set of known shapes selected for a given set of parameter constraints?
* How sensitive is the method to the choice and number of known shapes?
* Does the method require the SDF networks to be trained in a specific way?
* Would incorporating the linearity-mismatch metric into equation (1) as a regularization term improve the quality of the generated shapes?

---

> ### Author Response · Authors · 2025-11-20
>
> We thank the reviewer for the thoughtful and constructive feedback. Below we address all weaknesses and questions in detail.
>
> **Q1. How are the set of known shapes selected for a given set of parameter constraints?**
> LAMP does not require manual exemplar selection. All exemplars contribute to the linear system:
>
> $ \min_\alpha \|P^\top_{:,C}\,\alpha - p_{d,C}\|_2^2 \quad \text{s.t.} \quad 1^\top \alpha = 1. $
>
> The solution automatically identifies the most relevant shapes by assigning a corresponding weight to each exemplar, producing an $\alpha$ vector in parameter space that is then applied directly to the aligned SDF decoder weights to generate the target geometry.
>
> **Q2. How sensitive is the method to the choice and number of known shapes?**
> This is an important point, and our ablation study (Appendix E) directly measures this sensitivity. We find that increasing the number of exemplars consistently expands the safe extrapolation range, while sparse coverage leads to higher linearity-mismatch values, allowing the safety metric to reliably flag unreliable outputs.
> As shown in Table 7 (Appendix E), increasing the number of known shapes from 10 to 100 samples expands the mean safe single-parameter extrapolation range from about 146% to 331%. This more than 2× increase illustrates that when the exemplar shapes cover the parameter space more uniformly, LAMP preserves local linearity much more effectively, leading to substantially better controllability and reliability.
>
> **Q3. Does the method require the SDF networks to be trained in a specific way?**
> Yes. LAMP requires all SDF decoders to be overfit starting from a common initialization, where this initialization is simply the fully trained SDF network of one exemplar shape. Using this shared starting point keeps all subsequent overfit networks within the same local basin of the weight space, which is essential for Assumption A2 to hold. We include an ablation study justifying this initialization in Appendix F.
>
> **Q4. Would adding the linearity-mismatch metric as a regularizer improve generation?**
> Incorporating the linearity-mismatch metric directly into Eq. (1) as a regularizer would, in principle, improve reliability, but computing the safety metric at every iteration requires running the $N$ unflattened SDF decoders on all 3D query points, which is computationally expensive. We instead use the norm of the mixture coefficients $\|\alpha\|_2$ as a lightweight proxy. This quantity varies monotonically with extrapolation and correlates strongly with the final safety score, making it an efficient indicator of how far a solution moves from the valid linear regime.
>
> To verify this idea, we implemented the following multi-objective optimization formulation:
>
> $\min_{\alpha}\ \|P_{:,C}^\top \alpha - p_{d,C}\|_2^{2} + \lambda \|\alpha\|_2^{2}, \qquad 1^\top \alpha = 1.$
>
> Because LAMP’s optimization is extremely fast, sweeping $\lambda$ allowed us to efficiently trace the Pareto frontier between parameter accuracy and geometric reliability. In practice, we observed predictable and smooth tradeoffs: larger $\lambda$ produces more conservative shapes with lower extrapolation, while smaller $\lambda$ favors accuracy but increases the risk of geometry deformation. We report the results of this multi-objective experiment and our observations in Appendix M of the revised manuscript.

---

> > ### Comment · Reviewer_CbJm · 2025-11-26
> >
> > Thank you for carefully adressing my questions.
> >
> > After reading the other reviews and responses, I maintain my position that this paper proposes a novel, controllable, efficient and low-data method that is a useful contribution to the community, despite its limitations regarding the geometric and topological variablity.

---

### Official Review · Reviewer_BDet · 2025-10-31

**Soundness:** 3
**Presentation:** 3
**Contribution:** 3
**Rating:** 6
**Confidence:** 3

**Summary:**

This work proposes a framework for generating new 3D shapes given a set of physical parameters. The key idea is to first fit an SDF-based decoder (similar to DeepSDF) for each training shape, using a dataset of shapes with annotated physical parameters. To generate a shape with target parameters, the method creates an affine combination of decoder network weights, mixed according to the parameters of the training set. The resulting network weight is then used to decode the mesh. Experimental results on DrivAerNet++ and BlendedNet show that the framework achieves better prediction accuracy for interpolation and better generalization for extrapolation. Additionally, a safety metric is proposed to avoid invalid generation results, and an application for drag reduction demonstrates effectiveness for aerodynamic optimization.

**Strengths:**

- To my knowledge, existing work mainly focuses on generating network weights or latents, such as HyperDiffusion. The idea of interpolating network weights with shared initialization to obtain new generation results is both interesting and novel.
- Despite its simplicity, the experimental results demonstrate that the framework achieves better performance on both DrivAerNet++ and BlendedNet datasets for interpolation and extrapolation. The framework also enables aerodynamic optimization and achieves better optimization results than the baseline.
- The proposed safety metric is shown to be effective both quantitatively and qualitatively in Figures 5 and 17–18.

**Weaknesses:**

The current manuscript has some minor concerns:

- This work assumes generated 3D shapes are produced from an affine transformation of the input shapes (Theorem A(1)). The proposed framework is unlikely to work on datasets with heterogeneous topologies. It is recommended to discuss this assumption and limitation.
- Theorem A(2) assumes that if the weight difference from initialization is small enough, the obtained mixed SDFs will be close to a linear combination of the original SDFs. While this sounds reasonable theoretically, an ablation study would help justify the necessity of using a shared initialization.

**Questions:**

- Could the proposed framework work on datasets with heterogeneous topologies (e.g., shapes with varying numbers of holes or disconnected components)? If not, could you expand the discussion on this limitation and clarify the scope of applicability?
- Could you provide an ablation study comparing the performance of LAMP with and without shared initialization? This would help justify the necessity of the shared initialization assumption stated in Theorem A(2).

---

> ### Author Response · Authors · 2025-11-20
>
> We thank the reviewer for the constructive feedback and address each point below.
>
> **Q1. Could the framework work on datasets with heterogeneous topologies?**
> We agree that LAMP, in its current form, does not naturally extend to datasets with heterogeneous topologies. This limitation follows directly from **Assumption A1**, which requires that geometric variation across exemplars can be captured by a shared linear control-point structure. When topology varies substantially (e.g., differing numbers of holes or disconnected components), a single linear feature basis cannot represent all shapes consistently, and affine weight-space mixing may become ill-posed.
>
> We clarified this limitation in the revised manuscript and specify that LAMP is best suited for parametrically structured shape families with consistent topology, such as shapes sharing the same number of holes or connected components. We included an additional example in appendix O demonstrating successful mixing on a geometry with a hole, where all exemplars share the same topological structure.
>
> Moreover, the framework could be extended to heterogeneous-topology datasets by grouping shapes into topology-consistent classes based on discrete structural variables (e.g., number of holes, presence or absence of add-on parts), and applying LAMP independently within each class.
>
>
> **Q2. Could you provide an ablation comparing LAMP with and without shared initialization?**
>
> We agree with the reviewer that an ablation study is important to justify Assumption A2. We have performed this experiment and included the results in Appendix F of the revised manuscript.
>
>
>
> When using a shared initialization, the exemplar SDF networks remain well aligned in weight space. Affine mixing under this setup produces plausible and parameter-consistent shapes, and the linearity mismatch remains low and stable across the dataset. These observations indicate that the shared initialization keeps the networks within a locally linear region of the weight manifold where mixing behaves predictably.
>
> In contrast, removing the shared initialization causes each network to fall into a different basin of attraction. The resulting weight spaces become misaligned, and mixing their weights produces invalid outputs, including severe distortions, collapsed shapes, and non-car-like geometry. This clear degradation demonstrates that alignment is lost without a shared starting point.
>
> To quantify this, we report R², MAE, and safe extrapolation range for single-parameter extrapolation sweeps using 100 exemplars:
>
> | **Condition**         | **R²** | **MAE** | **Mean Safe Extrapolation Range** |
> |------------------------|--------|---------|-----------------------------------|
> | **Shared Initialization**          | 0.838  | 0.507   | 330%                             |
> | **No Shared Initialization**  | -37.77 | 9.16    | 0%                               |
>
> These findings directly support the necessity of a shared initialization for maintaining local linearity in weight space and empirically validate Assumption A2.

---

> > ### Comment · Reviewer_BDet · 2025-11-28
> >
> > The additional experiments and clarifications have addressed my issue. I will maintain the current rating.

---

### Official Review · Reviewer_GHuJ · 2025-11-05

**Soundness:** 3
**Presentation:** 3
**Contribution:** 3
**Rating:** 8
**Confidence:** 4

**Summary:**

This paper proposes LAMP, a data efficient linear affine weight space model for 3D shape generation and interpolation/extrapolation that supports parameter control. Data scarcity is a major practical challenge, especially for engineering designs. The method is overall plausible and the paper shows extensive evaluation.

**Strengths:**

The proposed method provides a data-efficient framework for parameter-controlled 3D mesh generation.

The paper demonstrates the effectiveness of the method on some benchmark datasets.

**Weaknesses:**

Although the safety metric is shown to be effective for the test case (when compared with human judgement), it lacks theoretical guarantee why a simple threshold is sufficient.

The paper should discuss limitations/failure cases more clearly.

The process from SDF to mesh may lead to changing mesh connectivity even when the parameters only change by a small amount.

**Questions:**

How do the parameter constraints work in practice, especially when some constraints may have conflict to be satisfied at the same time?

---

> ### Author Response · Authors · 2025-11-20
>
> We thank the reviewer for the positive assessment and the insightful comments. We address the concerns and questions below.
>
> **Safety metric and theoretical guarantees**
>
> The safety metric directly measures how well our linearity assumption (A2) holds for a given affine weight combination. A high linearity mismatch indicates departure from the locally linear region where weight mixing behaves predictably.
>
> Because the metric is grounded in deviation from the idealized linear SDF relationship, the threshold can be strict or relaxed depending on the application’s tolerance for geometric distortion. In aerodynamic design studies, for instance, we use a conservative cutoff, whereas more exploratory generative settings may allow larger mismatch. We made this explicit in the revised paper.
>
> We also agree that the manuscript should describe limitations more clearly. We added a dedicated limitation section outlining the main situations where LAMP may fail. These issues arise primarily under far extrapolation, where the mixed weights leave the locally linear region assumed in A2. Additional failure modes include cases where exemplar coverage is insufficient, parameter constraints are mutually conflicting, topology differs across exemplars, or highly localized geometric features vary nonlinearly and are not well captured by affine weight mixing.
>
>
> **Q: How do the parameter constraints work in practice, especially when some constraints may have conflict to be satisfied at the same time?**
>
> In practice, parameter constraints in LAMP are enforced through a linear system that solves for mixing coefficients satisfying the desired physical and performance parameters simultaneously. When these constraints are mutually conflicting (e.g., requesting a geometry with a physical feature typically associated with high drag while enforcing a low drag coefficient), LAMP still computes a feasible affine combination of exemplar weights that best fits all targets.
>
> However, when decoding and meshing the corresponding unflattened SDF, the resulting geometry often exhibits visible deformation, an indication that the requested parameter combination lies outside the valid linear regime. In such cases, the linearity-mismatch safety metric reliably flags the generation as invalid. This situation can also be handled using a multi-objective formulation, where one jointly optimizes parameter satisfaction and an extrapolation proxy such as $||\alpha||$, which correlates strongly with linearity mismatch. Because LAMP’s optimization is extremely fast, exploring these trade-offs for conflicting constraints is computationally inexpensive and allows users to obtain feasible solutions that balance accuracy and geometric reliability.
>
> We included this discussion in Appendix M of the revised manuscript, along with visual examples illustrating how conflicting parameter constraints lead to detectable geometric distortions captured by the safety metric.

---

### Official Review · Reviewer_MKVD · 2025-11-05

**Soundness:** 3
**Presentation:** 3
**Contribution:** 2
**Rating:** 4
**Confidence:** 3

**Summary:**

This paper proposes LAMP, a data-efficient framework for parameter-controlled 3D mesh generation. The method employs affine mixing in the weight space of neural signed distance function (SDF) decoders to construct a generative model. Each exemplar shape in a small-scale dataset is fitted with an individually overfitted SDF decoder. For novel shape synthesis, mixing coefficients satisfying target parameter constraints are solved and then applied to linearly combine the exemplar decoders' weights. A linearity mismatch metric is introduced to detect and filter invalid geometries.

**Strengths:**

1. The paper is well-written, with clear and coherent presentation.
2. LAMP enables precise parameter-controlled generation with limited training data.

**Weaknesses:**

1. For each example, is the weight w the flattened vector of all parameters of its trained SDF network? What is the dimensionality D of w? Storing such large vectors per example may impose substantial storage costs—how does the method scale as the dataset grows?
2. Training a separate SDF network per example could be time-consuming. Could you report the overall training cost and per-example training time, and compare both training and inference time against the baseline methods.
3. The selection of baseline methods appears limited, with only two relatively early works (from 2019 and 2021). Could the authors clarify why these particular baselines were chosen instead of more recent ones? Would it be possible to include comparisons with newer approaches, such as modern conditional diffusion models fine-tuned on similarly small datasets?
4. Can the proposed method be integrated with recent SOTA 3D generative models? If not, does this limit its practical applicability or generalization potential?
5. Can the two linear assumptions proposed in the paper be supported by concrete feature data?
6. The provided code lacks the filtered_drivaernet_small.csv file and possibly the required mesh files, making it currently impossible to run. Why were these not included in the supplementary materials? Their absence may affect the transparency and reproducibility of the work.

**Questions:**

Please refer to weakness

---

> ### Author Response · Authors · 2025-11-20
>
> We thank the reviewer for the detailed and helpful questions. We address them point by point below.
>
> **For each example, is the weight w the flattened vector of all parameters of its trained SDF network?**
> Yes. For each exemplar, w is the flattened vector of all learnable parameters.
>
> **What is the dimensionality D of w?**
> D = 2,128,897.
>
> **Storing such large vectors per example may impose substantial storage costs—how does the method scale as the dataset grows?**
> Storage grows linearly with the number of exemplars \(N\):
>
> - One flattened decoder: ≈ 8.5 MB
> - 100 exemplars: ≈ 850 MB
> - 1000 exemplars: ≈ 8.5 GB
>
> This is still significantly more storage-efficient than storing meshes directly.
> A single high-resolution watertight mesh (e.g., from DrivAerNet++) typically requires 60-70 MB, so storing 100 meshes alone exceeds 6 GB, already larger than storing 100 LAMP decoders. In addition, decoder weights are resolution-free: they encode continuous SDFs, whereas meshes must be stored at a fixed (and often high) resolution.
>
> LAMP also supports fully online expansion of the exemplar bank. If new shapes or additional parameter variations become available later, one can simply overfit an SDF decoder to each new mesh and append it to the existing weight bank. This allows new geometric degrees of freedom, and even entirely new parameters, to be incorporated without retraining any global model. Most generative baselines cannot do this: adding new shapes or parameters typically requires full model retraining or fine-tuning.
>
> Finally, scalability at inference time is excellent. LAMP scales linearly with the number of exemplars, and solving for the affine mixing coefficients takes less than 10 ms. As shown in our ablation study (“How Does Sample Size Affect Reliability and Extrapolation in LAMP?”, Appendix E), increasing the number of exemplars improves both reliability and the safe extrapolation range, making the method robust as the dataset grows.
>
> **Is training one SDF network per exemplar expensive? How does it compare to baselines?**
> No, per-exemplar SDF overfitting is lightweight and fast.
>
> - LAMP:
>    - Per-shape SDF training:
>     ~5 minutes = 0.083 GPU-hours
>    - For 100 exemplars:
>   100 × 0.083 ≈ 8.3 GPU-hours
>   - Inference:
>     - Solving for $\alpha$: < 10 ms
>     -   SDF to mesh: < 5 s
>
> - DNI:
>   - SDF overfitting: ~8.3 GPU-hours
>   - DNI training: ~0.1 GPU-hours
>   - Inference: < 10 ms
>
> - AE-LPA:
>   - Training: ~10 GPU-hours
>   - Inference: < 10 s
>
> - Modern diffusion models (e.g., LION):
>   - Training: ~550 GPU-hours for about 2500 samples
>   - Inference: ~30 s
>   - Output resolution: only 2048 points, far lower than LAMP’s high-resolution SDFs (trained with 1M points per mesh) [1].
>
> Because LAMP computes per exemplar, it remains highly efficient in the small-data regime (10-100 shapes) and avoids the prohibitive cost of global generative model training.
>
> [1] Vahdat, Arash, et al. "Lion: Latent point diffusion models for 3d shape generation." Advances in Neural Information Processing Systems 35 (2022): 10021-10039.

---

> > ### Author Response · Authors · 2025-11-20
> >
> > **Could the authors clarify why these particular baselines were chosen instead of more recent ones?**
> >
> > We use DNI and AE-LPA as baselines because they are the only prior methods that share LAMP’s core ingredients: linear structure in weight or latent space and compatibility with low-data, parameter-controlled settings. DNI mixes directly in weight space, while AE-LPA enforces latent linearity, making them the most appropriate comparators.
> >
> > Modern 3D diffusion and generative models (e.g., SDFusion, GET3D, LION) require $10^3$-$10^5$ shapes and significant compute to function reliably. In the regime targeted by LAMP, 10-100 exemplars with explicit parametric control, these models tend to collapse, memorize, or ignore conditioning signals entirely. They also lack native parametric control and often produce outputs too coarse for engineering use. For example, LION generates only 2048-point clouds, far too low-resolution for detailed DrivAerNet geometry, and still requires ~550 GPU-hours per class to train despite this coarse output. [1]
> >
> > We further experimented with Hunyuan 3D [2], a state-of-the-art high-resolution 3D generator, by attempting to map our parameters into its latent space. Although Hunyuan 3D [2] offers strong unconditional quality and high reconstruction accuracy on DriverNet cars, learning a parameter to latent mapping proved ineffective. Using 1,000 parameter-shape samples, linear and nonlinear regressors achieved only R² scores between -0.1 and 0.3, showing that the latent space contains insufficient parametric structure to support meaningful control. Moreover, training a nonlinear mapping to predict latent vectors from parameters resulted in the loss quickly plateauing, confirming that even with substantially more data than the low-data regime used in LAMP, the latent space does not provide the disentangled, directionally consistent structure required for precise geometric parameterization.
> >
> > These findings underscore why modern diffusion models cannot serve as meaningful baselines for the parametric, small-data setting addressed by LAMP.
> >
> >
> >
> > **Can the proposed method be integrated with recent SOTA 3D generative models? If not, does this limit its practical applicability or generalization potential?**
> >
> > No, our method serves a different purpose than recent 3D diffusion models. LAMP is designed for precise parametric control, safe extrapolation, and efficient design-space exploration, whereas diffusion models (e.g., SDFusion, GET3D, LION) focus on unconditional or weakly-conditional generation, require large datasets (>100 samples), and cannot enforce exact parameter constraints. In contrast, LAMP achieves high-accuracy control with 100 samples and enables exploration of a design space ~8000× larger than the dataset volume when extrapolating by only 50% (Appendix N of the revised paper).
> > LAMP is well suited for engineering design loops, where practitioners repeatedly modify geometry, mesh, simulate, and validate. This cycle is slow even for small parameter changes. LAMP provides instant geometric and performance feedback for arbitrary parameter combinations, enabling rapid exploration before committing to simulation. Its interpretable parameters, online extensibility, and safety-aware extrapolation support practical, iterative aerodynamic and structural design workflows.
> >
> > **Can A1 and A2 be validated with concrete feature data?**
> >
> > We validate A1 and A2 with concrete feature data
> >
> > - **A1 (parameter to geometry linearity)**:
> >   In Figures 13-14, we show a strong linear mapping between geometric features extracted from point clouds (via a fixed PointNet encoder) and the associated design parameters (LASSO regressor, $R^2 > 0.9$).
> >
> > - **A2 (linearity in SDF weight space)**:
> >   We compare  $ f(z;\sum_i \alpha_i w_i) \text{ vs. } \sum_i \alpha_i f(z;w_i) $
> >   across ten thousand query points. The relationship is highly linear in-distribution (R² ≈ 0.99) and decays with extrapolation. This plot was added in Appendix K of the revised manuscript.
> >
> >
> >
> > **Missing files**
> > All referenced meshes and metadata are part of the public DrivAerNet++ and BlendedNet datasets. We will include the missing CSV, provide the mesh files (and download instructions), release the trained SDF decoders, and add an end-to-end runnable example in the final version to ensure full reproducibility.
> >
> > [1] Vahdat, Arash, et al. "Lion: Latent point diffusion models for 3d shape generation." Advances in Neural Information Processing Systems 35 (2022): 10021-10039.
> >
> > [2] Zhao, Zibo, et al. "Hunyuan3D 2.0: Scaling Diffusion Models for High Resolution Textured 3D Assets Generation." CoRR (2025).

---

### Official Review · Reviewer_peNY · 2025-11-11

**Soundness:** 2
**Presentation:** 2
**Contribution:** 2
**Rating:** 4
**Confidence:** 3

**Summary:**

This paper proposes LAMP (Linear Affine Mixing of Parametric shapes), a framework for data-efficient and controllable 3D mesh generation. Instead of training a single large model, LAMP aligns a small number of SDF (Signed Distance Function) decoders—each overfitted to one exemplar—into a shared weight space, and generates new geometries by solving a parameter-constrained affine mixing problem over these aligned weights. The paper introduces a linearity-mismatch safety metric to detect invalid extrapolations and demonstrates applications on two 3D design benchmarks (DrivAerNet++ and BlendedNet). Experimental results show improved interpolation, safe extrapolation, and physics-guided optimization with limited data compared to AE-LPA and Deep Network Interpolation (DNI).

**Strengths:**

Data-efficient and interpretable approach for parameter-controlled 3D generation. Clear problem motivation in engineering design, emphasizing low-data and extrapolation regimes. Practical safety check (linearity mismatch) that can identify unstable generations without ground truth. Demonstrates reasonable quantitative and visual improvements over strong baselines (DNI, AE-LPA). Readable and systematic experimental section, showing controlled interpolation, extrapolation, and performance optimization.

**Weaknesses:**

The main weakness of this paper lies in its limited conceptual novelty. The core idea of affine weight mixing closely follows earlier works like Deep Network Interpolation (DNI) and model interpolation methods, without introducing much theoretical advancement. The key assumptions behind the approach, especially (A1) and (A2), feel heuristic and are never really tested beyond small, local cases—there’s no solid evidence that SDF weights behave linearly when pushed far beyond the training range. The proposed safety metric is interesting but remains entirely empirical, with no proof or deeper justification linking linearity mismatch to actual geometric validity. Overall, the paper leans heavily on engineering experiments rather than theoretical grounding or broader applicability. It doesn’t discuss failure cases, scalability to more complex architectures, or how the method might fit into real-world design systems. Finally, the comparison set feels a bit dated—modern 3D diffusion or generative models like SDFusion, GET3D, or LION aren’t seriously considered, which weakens the broader relevance of the results.

**Questions:**

How sensitive is the method to the choice of initialization during SDF overfitting? Would random seeds break alignment?
Could the authors quantitatively evaluate the linearity assumption (A2) by plotting true vs. predicted SDF outputs under linear weight interpolation?
Could this framework be extended to diffusion-based decoders or non-implicit representations (e.g., mesh-based networks)?

---

> ### Author Response · Authors · 2025-11-20
>
> We thank the reviewer for the thoughtful and detailed feedback. Below we address the concerns and questions directly.
>
> **Conceptual novelty relative to DNI and model interpolation.**
> Our approach differs fundamentally from DNI-style methods: rather than learning a parameter-to-weight mapping, we build an aligned basis of overfit SDF decoders and perform constrained affine optimization directly in weight space. To our knowledge, no prior work formulates parameter-controlled 3D generation in this manner, nor enables multi-way constrained mixing and large-range extrapolation.
>
> **Failure cases.**
> Failures occur mainly under far extrapolation, insufficient exemplar coverage, conflicting parameter constraints, or inconsistent topology. In such cases, the safety metric reliably detects and suppresses invalid outputs. We updated the paper with a concise limitations section.
>
> **Scalability and incremental use.**
> LAMP scales linearly in the number of exemplars, and the affine solve takes <10 ms. Our sample-size ablation (Appendix E) shows that more exemplars increase reliability and safe extrapolation range. The method is also fully incremental. If new shapes or additional parameter variations become available months later, one can simply overfit an SDF decoder to each new mesh and append it to the existing weight bank. This allows new geometric degrees of freedom, and even entirely new parameters, to be incorporated without retraining any global model. Most generative baselines cannot do this: adding new shapes or parameters typically requires full model retraining or fine-tuning.
>
> **Applicability to design workflows.**
> LAMP is well suited for engineering design loops, where practitioners repeatedly modify geometry, mesh, simulate, and validate. This cycle is slow even for small parameter changes. LAMP provides instant geometric and performance feedback for arbitrary parameter combinations, enabling rapid exploration before committing to simulation. Its interpretable parameters, online extensibility, and safety-aware extrapolation support practical, iterative aerodynamic and structural design workflows.
>
> **Baselines and relation to modern 3D generative models.**
> We selected DNI and AE-LPA because they are the only prior methods that (i) rely on linear structures in weight or latent space and (ii) operate effectively in low-data regimes (10–100 exemplars), which directly matches the setting addressed by LAMP. In contrast, modern 3D diffusion and generative models such as SDFusion, GET3D, and LION require 10³–10⁵ training shapes and substantial computational resources to function reliably. In the low-data regime targeted by our method, these models tend to collapse, memorize, or ignore conditioning signals, and none provide native parametric control or reliable extrapolation. Their output resolution is also insufficient for engineering use: for instance, LION produces only 2048-point clouds, far too coarse for detailed DrivAerNet geometries, while still requiring ~550 GPU-hours per class to train [1]. These limitations make diffusion models unsuitable baselines for the N = 100 exemplar setting we consider.
>
> We also tested Hunyuan 3D [2], a state-of-the-art high-resolution generator, by attempting to map our parameters into its latent space. Although Hunyuan 3D [2] achieves strong unconditional quality and high reconstruction fidelity on DriverNet cars, mapping parameters to latents proved ineffective: with 1,000 samples, linear and nonlinear regressions achieved only R² between -0.1 and 0.3. This further confirms that even modern diffusion models do not provide the disentangled, directionally consistent latent structure needed for precise parametric control in low-data regimes.
>
> [1] Vahdat, Arash, et al. "Lion: Latent point diffusion models for 3d shape generation." Advances in Neural Information Processing Systems 35 (2022): 10021-10039.
>
> [2] Zhao, Zibo, et al. "Hunyuan3D 2.0: Scaling Diffusion Models for High Resolution Textured 3D Assets Generation." CoRR (2025).

---

> > ### Author Response · Authors · 2025-11-20
> >
> > **Q1. How sensitive is the method to the choice of initialization during SDF overfitting?**
> > We trained exemplar SDF decoders both *with* and *without* a shared initialization. With a shared initialization, all exemplars remain in a common local basin and LAMP generates stable, parameter-consistent shapes. Without it, the networks fall into different basins, weight-space alignment collapses, and affine mixing produces severely distorted, non-car-like geometries. This confirms that a shared initialization is essential for maintaining a coherent basis. We included this ablation and visual comparisons in Appendix F of the revised manuscript.
> >
> > **Q2. Would random seeds break alignment?**
> > We fine-tuned exemplar SDFs using different random seeds while keeping the shared initialization fixed. Alignment remained stable across all runs, generated shapes were visually similar to the same-seed baseline, parameters remained consistent, and A2 linearity held.
> >
> > To quantify this, we report R², MAE, and safe extrapolation range for single-parameter extrapolation sweeps using 100 exemplars:
> >
> > | **Condition**         | **R²** | **MAE** | **Mean Safe Extrapolation Range** |
> > |------------------------|--------|---------|-----------------------------------|
> > | **Same Seed**          | 0.838  | 0.507   | 330%                             |
> > | **Random Seed**        | 0.773  | 0.610   | 304%                             |
> > | **No Initialization**  | -37.77 | 9.16    | 0%                               |
> >
> > Using different seeds slightly reduces performance but does not break alignment. In contrast, removing the shared initialization destroys linearity entirely, leading to invalid generations. We included these ablations in Appendix F of the revised manuscript.
> >
> > **Q3. Could the authors quantitatively evaluate the linearity assumption (A2) by plotting true vs. predicted SDF outputs under linear weight interpolation?**
> > Yes. We computed linearity comparisons of  $ f(z;\sum_i \alpha_i w_i) $ vs. $ \sum_i \alpha_i f(z; w_i) $ over ten thousand 3D query points. The relationship is highly linear in-distribution (R² ≈ 0.99) and decays with extrapolation. This plot was added in Appendix K of the revised manuscript.
> >
> > **Q4. Could this framework be extended to diffusion-based decoders or non-implicit representations (e.g., mesh-based networks)?**
> > Yes. Theoretically, LAMP is architecture-agnostic: as long as all exemplar decoders are overfit from a shared initialization, affine weight mixing should still apply. To validate this, we ran an experiment with a point-cloud decoder, which produced smooth and coherent interpolations. However, SDF decoders remain the better choice for this application because they allow mesh extraction at arbitrary resolution. We included this experiment in Appendix L of the revised manuscript.

---

### Author Response · Authors · 2025-12-03
**Brief context for AC on reviewer consensus for LAMP**

Dear Area Chair,

We wanted to provide a short summary of the current reviewer landscape and how the discussion evolved, to complement the existing reviews and our rebuttal.
- **Overall split.** In the initial review, three reviewers were clearly positive (ratings 8, 8, and 6) and two were marginally below the acceptance threshold (4, 4) but indicated they would not mind if the paper is accepted.
- **Strongly positive reviews (8, 8).**
Reviewers GHuJ and CbJm view the paper as a useful contribution: they highlight the novelty of combining parameter vectors and neural SDF weights for controlled interpolation/extrapolation, the data‑efficient and controllable nature of the method, and its suitability for low‑data engineering settings. After reading our responses, Reviewer CbJm explicitly states that they maintain the position that the paper proposes “a novel, controllable, efficient and low‑data method that is a useful contribution to the community, despite its limitations regarding the geometric and topological variability.”
- **Positive review (6).**
Reviewer BDet finds the idea of interpolating SDF decoders from a shared initialization “interesting and novel,” and agrees that the framework performs well on both interpolation and extrapolation, with an effective safety metric. Their main concerns were about the topological assumptions and the role of shared initialization. We added explicit discussion of the topology issue and an ablation comparing with/without shared initialization (Appendix F). In a follow‑up comment, they note that the additional experiments and clarifications addressed their issue and that they will maintain their rating.
- **Borderline reviews (4, 4).**
Reviewers peNY and MKVD are marginally below acceptance, mainly due to concerns about (i) novelty relative to DNI/model interpolation, (ii) lack of modern diffusion‑based baselines, (iii) storage/training cost scaling, and (iv) empirical support for our linearity assumptions and safety metric. Our rebuttal and revisions directly address all these points, including:
  - Clarifying how LAMP differs from DNI and why it is tailored to low‑data parametric control rather than large‑scale diffusion models.
  - A detailed cost/scaling breakdown and storage comparison versus meshes and baselines.
  - New experiments validating A1/A2 (feature‑space and SDF‑space linearity), sensitivity to initialization and random seeds, and the behavior of the safety metric, as well as additional limitations discussion.

  While these two reviewers did not have a chance to update their ratings before the discussion window was abruptly closed, we believe all their main concerns are now addressed in the updated manuscript and appendices.

We hope this overview of the post‑rebuttal state, together with the full reviews, our detailed responses, and the new experiments, is helpful for your assessment. Thank you very much for your time and consideration.

---

### Note · Program_Chairs · 2026-01-17
**Submission Desk Rejected by Program Chairs**

The following references in this submission do not refer to real documents and/or have major errors in bibliographic information:

 Hang Chen and et al. Sdf-diff: Differentiable rendering of signed distance fields for 3d shape editing. In ECCV, 2022.
Yilun Chen and et al. Maskedgen: Conditional 3d shape generation with masked diffusion. 2023.
Ruida Du and et al. Cad2shape: Learning geometry representations from cad. In ICCV, 2023.
Hao Liu and et al. Gen-ldm: A generative latent diffusion model for 3d shape generation. arXiv preprint arXiv:2301.XXXX, 2023.